# Aspartate aminotransferase Rv3722c governs aspartate-dependent nitrogen metabolism in *Mycobacterium tuberculosis*

Robert S. Jansen [1], Lungelo Mandyoli[2], Ryan Hughes[2], Shoko Wakabayashi[3], Jessica T. Pinkham[3], Bruna Selbach[1], Kristine M. Guinn[3], Eric J. Rubin [3], James C. Sacchettini[2✉] & Kyu Y. Rhee [1,4✉]

Gene *rv3722c* of *Mycobacterium tuberculosis* is essential for in vitro growth, and encodes a putative pyridoxal phosphate-binding protein of unknown function. Here we use metabolomic, genetic and structural approaches to show that Rv3722c is the primary aspartate aminotransferase of *M. tuberculosis*, and mediates an essential but underrecognized role in metabolism: nitrogen distribution. Rv3722c deficiency leads to virulence attenuation in macrophages and mice. Our results identify aspartate biosynthesis and nitrogen distribution as potential species-selective drug targets in *M. tuberculosis*.

[1] Division of Infectious Diseases, Department of Medicine, Weill Cornell Medical College, New York, NY 10065, USA. [2] Department of Biochemistry and Biophysics, Texas A&M University, College Station, TX 77843-2128, USA. [3] Department of Immunology and Infectious Diseases, Harvard T. H. Chan School of Public Health, Boston, MA 02115, USA. [4] Department of Microbiology & Immunology, Weill Cornell Medical College, New York, NY 10065, USA.
✉email: sacchett@tamu.edu; kyr9001@med.cornell.edu

The accumulation of sequence data is outpacing the ability to annotate its information content[1]. As many as half of all predicted coding sequences are estimated to be unannotated or misannotated or to encode additional activities beyond their predicted functions[1]. Conversely, approximately one-third of all detected enzymatic activities lack an associated coding sequence[2]. For microbial pathogens, such gaps have restricted access to what may be the most specific and actionable features of their physiology.

*Mycobacterium tuberculosis* (*Mtb*) is the causative agent of tuberculosis (TB) and the leading cause of death due to an infectious agent[3]. Curative chemotherapies for TB were first developed over 50 years ago, but are only recently beginning to change, and currently number among the longest, most complex, and toxic treatments for a bacterial infection. Together, these shortcomings have fostered rates of treatment noncompliance and default that help fuel the pandemic and promote the emergence of drug resistance. Shorter, simpler chemotherapies are urgently needed.

*rv3722c* is an *Mtb* gene of unknown function, predicted by transposon mutagenesis to be essential for in vitro growth[4]. Bioinformatic analyses predict that *rv3722c* encodes a pyridoxal phosphate (PLP)-binding domain. A previously deposited crystal structure (PDB 5C6U [https://www.rcsb.org/structure/5C6U]) identified Rv3722c as a member of the class I family of PLP-binding proteins, also referred to as the aspartate aminotransferase family[5]. Despite this designation, members of the class I PLP-binding/aspartate aminotransferase protein families encode aminotransferases, ligases, lyases, decarboxylases, and even transcription factors[5,6]. Consistent with this diversity of functions, Rv3722c is annotated as an aminotransferase[7,8], a member of the GntR family of transcription factors[9,10], a serine hydrolase[11], and even a secreted protein[12]. Proteomics analysis also revealed that Rv3722c numbers among the top 10–25% of most abundant proteins in *Mtb*[13]. Together, the ambiguous annotation, essentiality, abundance, and PLP-binding domain, identify Rv3722c as a potentially attractive drug target.

Growing evidence has implicated carbon metabolism as a determinant of *Mtb* pathogenicity[14]. However, recent work has begun to implicate a similarly important role for nitrogen uptake and assimilation[15,16]. Here, we demonstrate that Rv3722c is the primary aspartate aminotransferase (AspAT) in *Mtb*, nonredundantly catalyzes the specific biosynthesis of Asp in vitro, and is essential for axenic growth and survival of *Mtb* in macrophages and in mice. We further show that this essentiality is due, in part, to a nonredundant role of Asp in the metabolic distribution of assimilated nitrogen. These findings identify Rv3722c as an essential metabolic mediator of Asp biosynthesis, and Asp-dependent nitrogen metabolism as an essential determinant of *Mtb* growth and virulence.

## Results

**Rv3722c is conditionally essential in vitro.** We first sought to confirm the predicted essentiality of Rv3722c. We constructed an *Mtb* strain in which expression of Rv3722c is regulated by its native promoter, but protein stability is controlled by a tetracycline-repressible protein degradation system (Rv3722c-TetON)[17]. An isogenic strain lacking a functional protein degradation system (Rv3722c-Control) served as a control. As predicted, omission of anhydrotetracycline (ATC) resulted in depletion of Rv3722c protein and attenuation of *Mtb* growth in Glu-based Middlebrook 7H9 medium (Fig. 1a, b). Slow and incomplete degradation of Rv3722c likely resulted in residual levels of Rv3722c that were sufficient to sustain slow growth, but below the experimental limit of detection by western blot, as

observed following withdrawal of ATC from Rv3722c-TetON in 7H9 media from day 6 onward (Fig. 1a, b). The growth defect could be overcome by culturing Rv3722c-TetON in the same medium supplemented with casein hydrolysate, which contains a complex mixture of amino acids and small peptides (Fig. 1c), and mitigated by culture on Middlebrook 7H10 agar (Fig. 1d). This conditional rescue made it possible to deplete Rv3722c to levels below the limit of detection (Fig. 1e) before a subsequent experimental challenge. Such predepletion completely prevented subsequent growth in unsupplemented 7H9 (Fig. 1f). Rv3722c could also be depleted below the experimental limit of detection by western blotting without impairing growth by culturing the cells in an Asn-based minimal medium (Sauton's) (Supplementary Fig. 1). Interestingly, we observed that Rv3722c-deficient *Mtb* grew like wild-type in Sauton's media supplemented with Glu at concentrations contained in 7H9, or in Sauton's media in which Asn was replaced with Glu, indicating that the growth attenuation observed in 7H9 media is not strictly dependent on Glu. We further observed that Rv3722c-deficient *Mtb* conversely failed to grow in 7H9 supplemented with Asn at concentrations contained in Sauton's, or in 7H9 in which Glu was replaced with Asn, indicating that the lack of a growth defect observed in Sauton's media is not strictly dependent on Asn.

**Rv3722c is essential for infection.** We next tested the essentiality of Rv3722c in mouse bone marrow-derived macrophages with Rv3722c-sufficient and Rv3722c pre-depleted *Mtb*. Growth of Rv3722c-deficient *Mtb* was compromised in both resting and interferon-γ-activated macrophages (Fig. 2a). In vivo testing in an aerosol infection model of TB in mice similarly revealed a striking attenuation of Rv3722c-deficient *Mtb* in lungs, as reported by a lack of growth (Fig. 2b) or their rapid and complete replacement by escape mutants that were no longer ATC-dependent (Supplementary Fig. 2).

**Rv3722c functions as an aminotransferase.** Based on bioinformatic evidence implicating Rv3722c as a PLP-dependent protein, we assayed purified recombinant Rv3722c for catalytic activity using activity-based metabolomic profiling (ABMP). ABMP is a biochemically unbiased method developed to "deorphan" unannotated metabolic enzymes that consists of incubation of a protein of interest with a highly concentrated homologous metabolite extract and monitoring for time- and protein-dependent changes indicative of catalysis by untargeted high-resolution mass spectrometry[18,19]. ABMP with Rv3722c identified several putative product features (fragments, adducts, dimers, and isotopes) that all co-eluted with and corresponded to a metabolite with m/z 144.03 [M–H]$^-$ (Fig. 3a). This metabolite was formed in a time- and Rv3722c-dependent fashion (Fig. 3b), and identified by mass spectral fragmentation as ketoglutaramate (keto-Gln), the keto acid of Gln (Fig. 3c).

Aminotransferases are the only known enzymatic source of keto-Gln[20]. We therefore tested Rv3722c for aminotransferase activity. Because aminotransferases catalyze the reversible reaction between amino acids and keto acids, we tested the ability of Rv3722c to generate $^{15}$N-labeled amino acids following incubation with amine-$^{15}$N-Gln and with the same unlabeled concentrated mycobacterial metabolite extract. This approach demonstrated a clear time- and protein-dependent formation of $^{15}$N-Asp and $^{15}$N-Glu, with matching depletion of αKG (Supplementary Fig. 3A). This approach conversely demonstrated depletion of Asp, His, Gln, and Trp upon addition of αKG (Supplementary Fig. 3B). These data thus annotate Rv3722c, at a class level, as an aminotransferase.

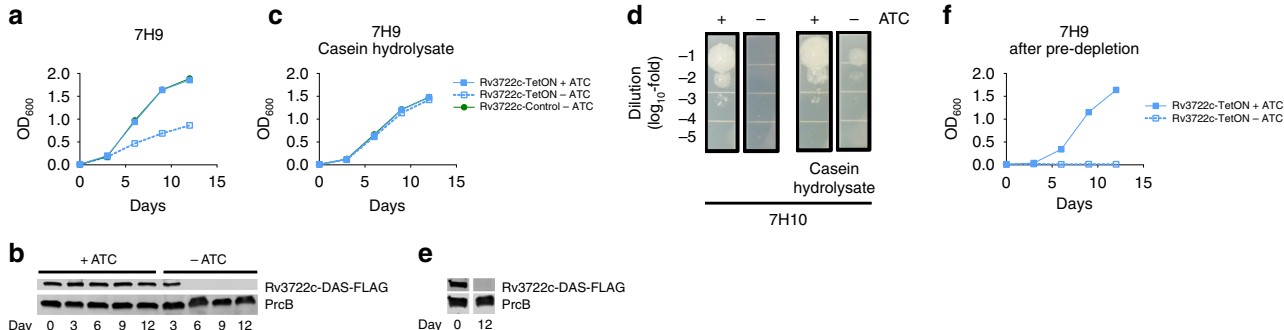

**Fig. 1 Rv3722c is conditionally esssential in vitro. a** Growth curve of Rv3722c-proficient and -deficient *Mtb* in 7H9 culture media. Rv3722c-TetON and Rv3722c-control were cultured in Middlebrook 7H9 culture media, with or without 500 ng mL$^{-1}$ anhydrotetracycline (ATC). Bacterial growth was monitored by optical density at 600 nm. **b** Western blot showing the depletion of Rv3722c 7H9 culture media. Rv3722c-TetON was cultured in 7H9 with or without ATC for 12 days (corresponding to **a**). Protein lysates were analyzed by western blotting, using an α-FLAG antibody. The proteasome subunit β (PrcB) was used as loading control. **c** Growth curve of Rv3722c-proficient and -deficient *Mtb* in 7H9 supplemented with casein hydrolysate. As **a**, but using 7H9 supplemented with 1% casein hydrolysate. **d** Spot assay of Rv3722c-proficient and -deficient *Mtb* on solid growth media. A serially diluted Rv3722c-TetON culture (OD 0.1) was spotted onto Middlebrook 7H10 agar with or without 1% casein hydrolysate, in the presence or absence of ATC, and cultured for 2 weeks. **e** Western blot showing depletion of Rv3722c. As **b**, but in 7H9 supplemented with 1% casein hydrolysate. **f** Growth curve of Rv3722c-proficient and -pre-depleted *Mtb* in 7H9 growth media. Rv3722c-TetON in 7H9 with or without ATC, after predepletion of Rv3722c in 7H9 with 1% casein hydrolysate without ATC. For all growth curves, data are represented as mean −/+ SD of three experimental replicates (*n* = 3) representative of at least two independent experiments. See also Supplementary Fig. 1. Source data are provided as a Source Data file.

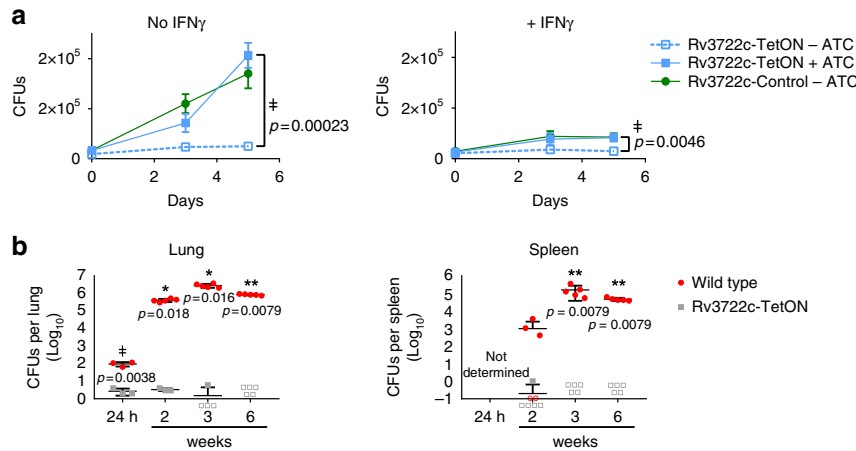

**Fig. 2 Rv3722c is required for virulence in macrophages and mice. a** Timecourse of macrophage infection. Primary murine bone marrow-derived macrophages were treated with control or interferon γ (IFNγ) for activation, followed by infection with Rv3722c-control and Rv3722c-TetON precultured in 7H9 + casein hydrolysate with or without ATC to predeplete Rv3722c (multiplicity of infection of one). The number of colony-forming units (CFUs) was assessed by plating serial dilutions on 7H10 solid media supplemented with casein hydrolysate and ATC. Data are presented as mean +/− SD of three experimental replicates (*n* = 3) representative of two independent experiments. **b** Timecourse of mouse infection. Mice were infected with aerosolized wild-type *Mtb* and Rv3722c-TetON precultured in Sauton's without ATC to generate Rv3722c-deficient *Mtb*. After infection, mice were fed chow without doxycycline. The numbers of CFUs in lung and spleen were assessed by plating serial dilutions on 7H10 solid media with and without ATC to determine the number of non-ATC-dependent mutants. No non-ATC-dependent Rv3722c-TetON mutants were detected at any timepoint. Data are represented as mean +/− SD of five mice (*n* = 5), except for the 24 h timepoints (*n* = 3) and Rv3722c-TetON 3 weeks in the lung (*n* = 4). Open symbols indicate samples in which the bacterial burden was below the limit of detection (<2 CFUs). A similar experiment is shown in Supplementary Fig. 2. Source data are provided as a Source Data file. Statistically significant differences were identified in panel **a** by a combination of a two-sided unpaired Students *t* test (*p* < 0.05, as indicated) and Mann–Whitney *U* rank testing (for which the smallest possible *p*-value with *n* = 3 is 0.1) (double dagger); and in panel **b** by an unpaired, two-sided Mann–Whitney *U* rank testing (**p* < 0.05, ***p* < 0.01, as indicated), except for the 24 h timepoint, in which statistically significant differences were determined as in **a**.

**Rv3722c functions as an aspartate aminotransferase**. To define the substrate specificity of Rv3722c, we next incubated Rv3722c with a panel of 26 amino acids, using αKG, pyruvate, and oxaloacetate as potential amino acceptors. Oxaloacetate and αKG, but not pyruvate, served as amino acceptors, while kynurenine (Kyn), Asp, and Glu served as the best amino donors (Supplementary Fig. 4A). In contrast, His, Cys, Asn, and Gln exhibited minimal activity (Supplementary Fig. 4A, B). Formal kinetic

studies identified Asp as the preferred substrate and Kyn as a weaker alternative (Fig. 3d), with affinities towards Asp that were similar to AspATs from other species[21,22]. His, Cys, Asn, and Gln exhibited extremely low rates of turnover consistent with kinetic side reactivity (Supplementary Fig. 4C). These results thus annotate Rv3722c as an aspartate aminotransferase (AspAT; Fig. 3e), with a weak, but significant, side activity toward Kyn, as observed in AspATs from other species[23].

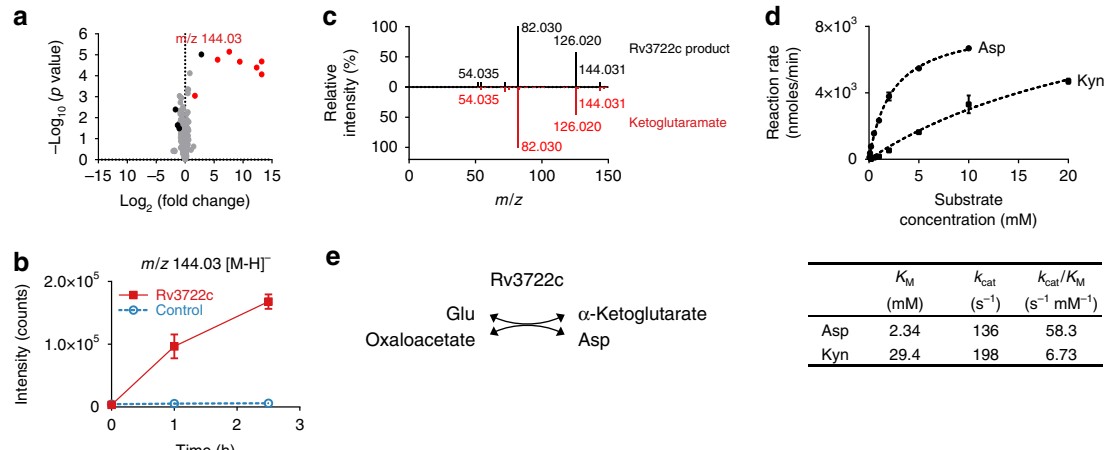

**Fig. 3 Rv3722c functions as an aspartate aminotransferase. a** Volcano plot of activity-based metabolite profiling with purified recombinant Rv3722c. Purified recombinant Rv3722c (10 μM) was incubated with a mycobacterial metabolite extract for 0 h or 2.5 h at 37 °C and analyzed using untargeted LC-MS. Each dot represents a feature (a chromatographic peak with a specific m/z) in the negative ionization mode; red dots represent features related (fragments, adducts, dimers, and isotopes) to the feature with m/z 144.03 [M–H]⁻, while black dots represent uncharacterized features with a fold change greater than two and p-value below 0.05 (n = 3). **b** Timecourse of Rv3722c-dependent formation of m/z 144.03. Same as **a**, but data shown for 0, 1, and 2.5 h, in the presence of active Rv3722c and heat-inactivated Rv3722c control (10 min 95 °C) (mean +/− SD of three experimental replicates (n = 3) representative of at least two independent experiments). **c** MS fragmentation spectra of the Rv3722c reaction product with m/z 144.03 and in-house synthesized ketoglutaramate at a collision energy of 10 V. **d** Steady-state enzyme kinetics of Rv3722c. Purified recombinant Rv3722c (0.01 μM) was incubated with 10 mM α-ketoglutarate and increasing concentrations of amino donors at 37 °C. Glu formation was measured by RapidFire mass spectrometry and used to determine initial reaction rates. The data was fitted to Michaelis–Menten kinetics using Graphpad Prism software, and are represented as mean +/− SD of three experimental replicates (n = 3) (Kyn: kynurenine). **e** Aminotransferase reaction catalyzed by Rv3722c. See also Supplementary Figs. 3 and 4. Source data are provided as a Source Data file.

**Rv3722c differs from classical aspartate aminotransferases.** Aminotransferases are assigned to five taxonomic classes based on the sequence of their PLP-binding domain[5,24]. AspATs belong to class I, which has classically been subdivided into types Ia and Ib[25]. Rv3722c, however, belongs to a recently described and structurally distinct subclass of AspATs, designated type Ic[25], that corresponds to the poorly characterized Pfam family PF12897[7]. Members of this family are absent in humans and almost exclusively present in bacteria, where they have a limited species distribution (Supplementary Fig. 5)[7].

**Rv3722c has a type Ic fold.** To elucidate the structural basis of substrate recognition by Rv3722c, we have solved structures of the enzyme co-crystalized with Glu or Kyn at a resolution of 2.6 Å and 2.23 Å, respectively (Supplementary Table 1 and Supplementary Note 1). The overall fold of Rv3722c is similar to that of other members of the recently described type Ic subgroup of PLP-binding proteins[25] (Supplementary Fig. 6A, B and Supplementary Note 2). Each monomer consists of a core domain (Pro52-Gly302) that folds into a nearly perfect α/β motif comprised a central eight-stranded (predominantly antiparallel) β-sheet surrounded by eight α-helices. Rv3722c also contains an N-terminal auxiliary domain (Ser2-Leu53 and Asp303-Ser430) that stacks and forms an elongated segment on top of the core domain, and consists of five α-helices and an antiparallel β-sheet.

**Rv3722c binds dicarboxylic acid and aromatic substrates.** The structure of Rv3722c pre-incubated with Glu demonstrated clear electron density of the amino acid in the active site located at the convergence of the auxiliary and core domain (Supplementary Fig. 7 and Supplementary Note 3). Glu is engaged by both polar and nonpolar interactions (Fig. 4a). Canonical AspATs coordinate the side-chain carboxylate group of dicarboxylic substrates through a conserved active site Arg (or Lys)[26,27]. Our structure shows that Rv3722c also forms a salt bridge with side chain

carboxylate group of Glu, but using a structurally non-homologous Arg residue at position 141.

In the asymmetric unit of Rv3722c pre-incubated with Kyn, we observed that the chains were bound to either the external aldimine intermediate PLP-kynurenine (PLP-Kyn), or the final keto acid product kynurenic acid (Kyna) (Fig. 4b, c; Supplementary Figs. 8A and 9 and Supplementary Note 4). In the unbound form of Rv3722c, the positively charged guanidinium side chain of Arg36 points outward and lies above the entrance of the active site pocket (Supplementary Fig. 10). Upon binding Kyn, however, we observed that the side chain of this residue is positioned more inward (by ~12 Å measured from the Cζ atom) and interacts with Kyn (Fig. 4b; Supplementary Figs. 8B and 10). Upon moving inward, Arg36 forms a bidentate salt bridge with Asp140. This ligand-induced conformational change is reminiscent of the so called "arginine switch" observed in type Ia AspATs, although here the Arg residue moves toward the substrate rather than away[28]. In the first-half reaction of transamination, a water molecule hydrolyzes the ketimine intermediate releasing a keto acid of the amino donor and generating PMP[29]. In our structure, we observed a sharp peak of a water molecule in close proximity to both the active site Lys257 and the PLP-KYN intermediate (Supplementary Fig. 8A). We speculate that this is the same water molecule responsible for the generation of the keto acid Kyna, from the PLP-KYN intermediate.

On the other hand, we observed that the product Kyna exhibits a different pose relative to its precursor. When compared with Kyn, the carboxylate moiety of Kyna is rotated by ~180° and faces the entrance of the solvent-exposed binding pocket (Fig. 4c). Kyna is stabilized mainly by nonpolar contacts in the binding pocket. The only major interaction stabilizing the product is a hydrogen bond between its hydroxyl group and NE group of Arg141. Interestingly, in all the Kyna-bound chains, Arg36 adopts its original "outward" conformation (Fig. 4c), most likely facilitating the release of the product.

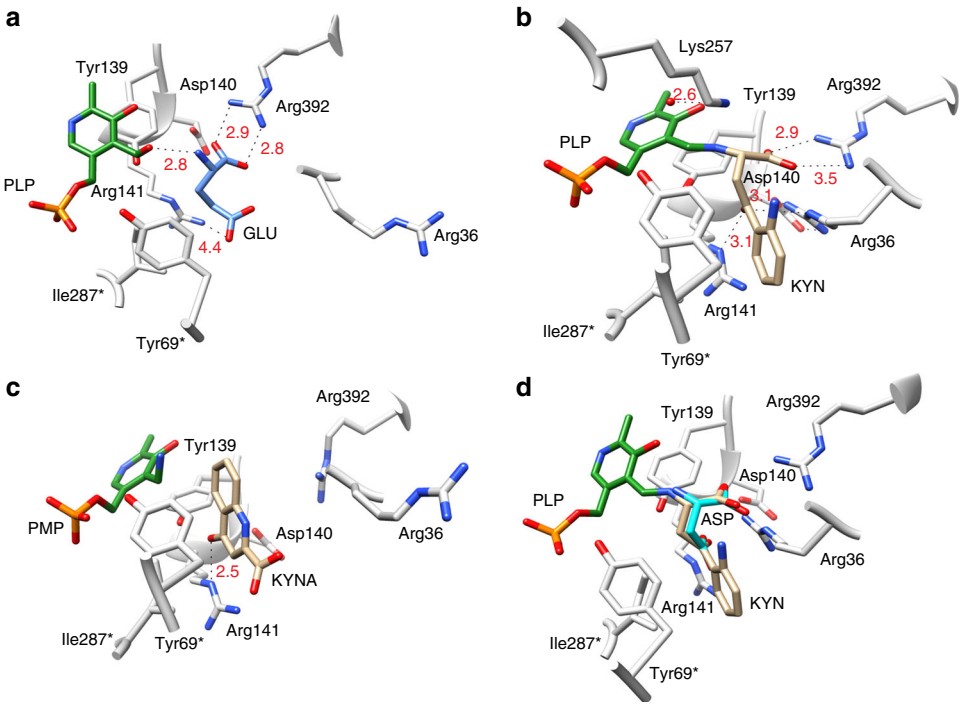

**Fig. 4 Rv3722c structure with bound ligands. a** Stick representation of Rv3722c in complex with glutamate (GLU; blue). **b** Rv3722c in complex with kynurenine (KYN; tan). **c** Rv3722c in complex with kynurenic acid (KYNA; tan). **d** Kyn mimics Asp in the active site pocket of Rv3722c. Modeling aspartic acid (ASP; cyan) into the electron density of the aliphatic chain of KYN (tan) reveals a structural mimicry that is potentially behind the capability of Rv3722c to use KYN as an amino donor. For all figures, active site residues are in light gray, and pyridoxal phosphate (PLP) or pyridoxamine phosphate (PMP) in green. Ligand interactions are shown as dashed lines, and distances are in angstroms. (Asterisk denotes a residue from the second monomer.) See also Supplementary Figs. 6–10.

**Kyn mimics Asp in the active site pocket of Rv3722c**. Attempts to solve the structure of Rv3722c bound to Asp or its analog α-methyl aspartate were unsuccessful. When Asp was superimposed with the aliphatic chain of Kyn, we not only observed that the amino groups and α-carboxylates overlapped but also that the carbonyl oxygen of Kyn is at a position analogous to the O6 atom of the carboxyl side chain of Asp (Fig. 4d). As a result, the carbonyl oxygen is well located to hydrogen bond with Arg141 and Arg36. In addition, the binding pose observed in Kyn-bound Rv3722c allows the aromatic ring to make hydrophobic contacts with nearby residues Tyr69* and Ile287*, further stabilizing the ligand prior to transamination. Our data thus show that the aliphatic backbone of Kyn mimics the dicarboxylic amino acid Asp, and provide a possible explanation why Rv3722c—and potentially other AspATs—display side activity towards the structurally disparate Kyn (see also Supplementary Note 5)[23,30].

**Rv0337c/AspC and Rv3565/AspB do not function as AspATs**. Despite the foregoing genetic, structural, microbiological, and in vitro biochemical evidence establishing Rv3722c as an essential AspAT, existing annotations of the *Mtb* genome include two additional AspATs (Rv3565/AspB and Rv0337c/AspC), one of which (Rv0337c/AspC) was also predicted to be essential for in vitro growth[4]. Unlike Rv3722c, both Rv0337c/AspC and Rv3565/AspB are annotated as class I AspATs, and belong to the 1a subclass[7,8,24]. We resolved this ambiguity by conducting ABMP on both enzymes. These studies demonstrated formation of keto-Gln and keto-Ile/Leu with Rv3565/AspB, and αKG with Rv0337c/AspC (Supplementary Fig. 11). ABMP in the presence of [15]N-labeled amino acids and their keto acids confirmed these activities (Supplementary Figs. 12 and 13). These results indicate that Rv0337c/AspC functions as an alanine aminotransferase

AlaT, while Rv3565/AspB functions as an alanine/valine aminotransferase AvtA with additional activity toward methionine. Both interpretations were confirmed by steady-state kinetic assays, which revealed substrate affinities similar to those from other species[31] (Supplementary Fig. 14).

**Rv3722c is the main AspAT in Mtb**. To test for other unannotated AspATs in *Mtb*, we traced the metabolic fates of [15]N-Asp and [15]N-Glu in Rv3722c-sufficient and -depleted *Mtb*, into [15]N-Glu and [15]N-Asp, respectively (Fig. 5a). In both cases, we observed strictly Rv3722c-dependent transfer of the labeled amino group to the corresponding keto acid, establishing that Rv3722c is the main AspAT in *Mtb*, potentially capable of running in both directions.

**Rv3722c balances anaplerosis and cataplerosis of the TCA cycle**. The conversion of Asp to oxaloacetate and Glu to αKG represent anaplerotic reactions of the TCA cycle, while their reverse reactions are cataplerotic. Rv3722c's ability to couple one anaplerotic half-reaction of the TCA cycle with a cataplerotic counterpart suggested that its activity might help enable balanced entry and exit into and out of the oxidative and reductive arms of the TCA cycle. To test this model, we incubated *Mtb* with [13]C-labeled Asp and Glu, and monitored for Rv3722c-dependent labeling of TCA cycle metabolites (Fig. 5b). We observed that Asp could serve as carbon source for the TCA cycle in a manner primarily dependent on Rv3722c and that anaplerosis from Glu could be suppressed in the absence of Rv3722c. These results thus demonstrate that Rv3722c appears poised to mediate anaplerosis from Asp and Glu while simultaneously facilitating cataplerosis of their corresponding partner keto acids.

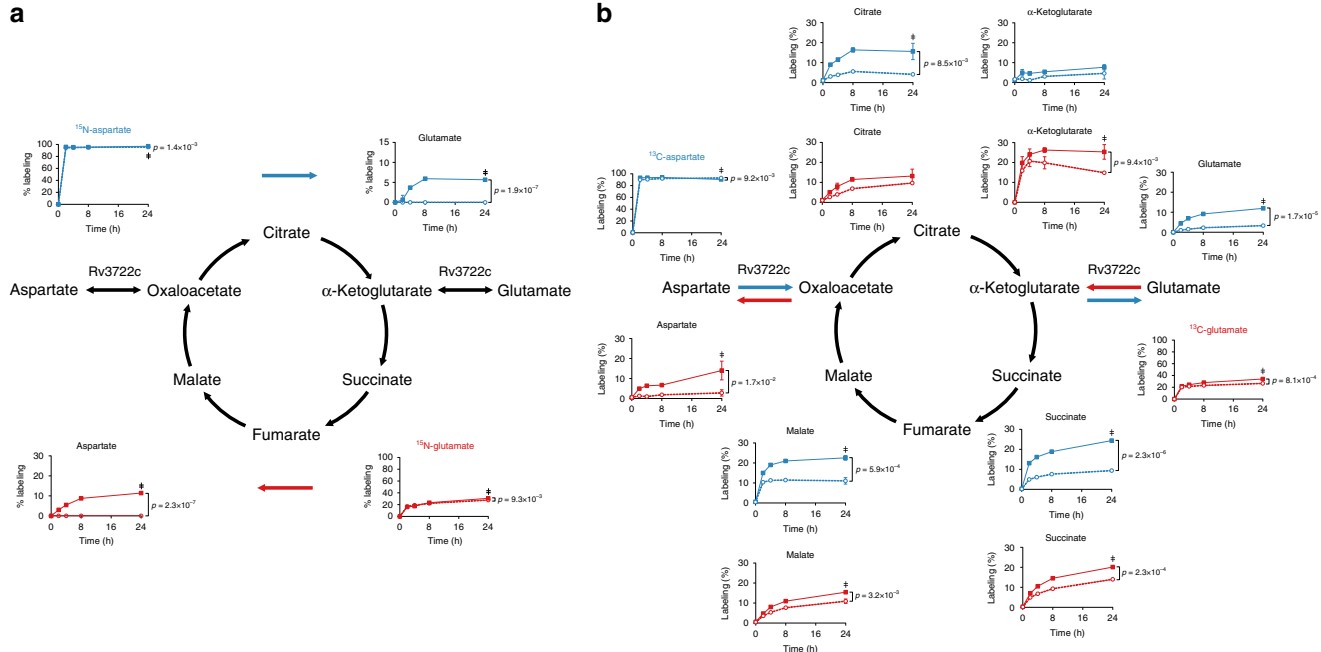

**Fig. 5 Rv3722c is the main aspartate aminotransferase in Mtb. a** [15]N-stable isotope tracing in Rv3722c-proficient and -deficient *Mtb*. Rv3722c-TetON was cultured in 7H9 supplemented with casein hydrolysate, with or without ATC, to predeplete Rv3722c. After a 3 h adaptation to unsupplemented 7H9 with (solid line) or without ATC (dashed line), [15]N-Asp (blue) or [15]N-Glu (red) was added to the media at a final concentration of 3 mM. After 0, 2, 4, 8, and 24 h at 37 °C, [15]N labeling of the indicated metabolites was determined using LC-MS. Colored arrows indicate the direction of the Rv3722c-mediated reaction. **b** [13]C-stable isotope tracing in Rv3722c-proficient and -deficient *Mtb*. Same as (**a**), but using [13]$C_4$-Asp (blue) or [13]$C_5$-Glu (red). Percentages are relative to the sum of all isotopes and corrected for natural isotope abundance. Data are presented as mean +/− SD of three experimental replicates ($n = 3$, but $n = 2$ for all 2h-ATC timepoints in **a** and Asp+ATC-4h, [15]N-Glu+ATC-4h in **a**, due to sample loss), representative of two independent experiments. Statistically significant differences were identified by a combination of a two-sided unpaired Student's *t* test ($p < 0.05$, as indicated) and Mann–Whitney *U* rank testing (for which the smallest possible *p*-value with $n = 3$ is 0.1) (double dagger); Source data are provided as a Source Data file.

**Rv3722c couples nitrogen assimilation to Asp synthesis.** Given the ability of Rv3722c to couple Glu and Asp biosynthesis to one another and the widely conserved role of Glu in primary assimilation of nitrogen, we sought to define the role of Rv3722c in coupling nitrogen assimilation to Asp biosynthesis. To do so, we exposed pre-depleted Rv3722c-TetON to a nitrogen up- and downshift, in a minimal medium with ammonia as sole nitrogen source[32]. In *E. coli*, nitrogen upshift causes a rapid decrease in αKG levels and slight increase in Glu levels[33]. We observed a similar, though less pronounced, effect in *Mtb*, which was not dependent on Rv3722c (Fig. 6). In Rv3722c-sufficient *Mtb*, the nitrogen-dependent increase in the Glu-αKG ratio resulted in an increase in Asp synthesis, while no Asp biosynthesis was observed in Rv3722c-deficient cells (Fig. 6). These results thus confirm that Asp synthesis is both strictly dependent on Rv3722c and coupled to Glu-mediated assimilation of inorganic nitrogen.

**Aspartate supplementation specifically rescues growth.** Our initial growth experiments demonstrated that Rv3722c was dispensable in the presence of casein hydrolysate, a complex mixture of amino acids and small peptides (Fig. 1c). We therefore sought to determine the specific amino acid(s) responsible for remedying the growth impairment of Rv3722c-deficient *Mtb* by culturing Rv3722c pre-depleted *Mtb* in 7H9 supplemented with each of 19 amino acids. Strikingly, only Asp was able to restore growth, linking Rv3722c's in vitro activity to its intrabacterial metabolic activity and growth (Fig. 7a).

**Aspartate is required for synthesis of essential metabolites.** The selective, though partial (Fig. 7b), rescue by Asp suggested that Rv3722c-deficient cells suffer from a growth-limiting deficiency

of Asp. Additional supplementation experiments showed that levels of the second Rv3722c reaction product, αKG (Fig. 3e), were not growth limiting (Supplementary Fig. 15). To confirm the Asp deficiency and explore its consequences on *Mtb* metabolism, we compared the metabolic profiles of Rv3722c-sufficient and -deficient bacteria, in the presence and absence of exogenous Asp. We focused on metabolites that are formed from Asp by enzymes that are essential for *Mtb*[4]. Figure 7c depicts the changes related to Asp and its essential downstream products, while the full metabolite profiles collected under these and other growth conditions are shown in Supplementary Fig. 16. In the absence of exogenous Asp, intracellular Asp levels were reduced approximately twofold in the absence of Rv3722c. Levels of most essential downstream Asp-dependent products, however, were depleted to a far greater degree than their precursor, Asp (Fig. 7c). Supplementation with Asp restored Asp pools to supraphysiologic levels with a similar accumulation of several downstream intermediates, such as nicotinic acid and homoserine, but corrected others, such as succinyl-DAP, less completely (Fig. 7c).

To identify the specific growth-limiting, Asp-dependent metabolites among those that could not be corrected by exogenous Asp, we supplemented Asp-containing media with individual downstream metabolites. No single metabolite could completely restore wild-type levels of growth. However, specific effects of pantothenate, hypoxanthine and Arg were observed in both supplementation and dropout screens (Fig. 7d, e).

**Rv3722c regulates Asp-dependent distribution of assimilated nitrogen.** Recognizing the ability of Asp to serve as a donor of both carbon and nitrogen[34], we sought to determine the specific role of Rv3722c in the distribution of nitrogen via Asp. To do so,

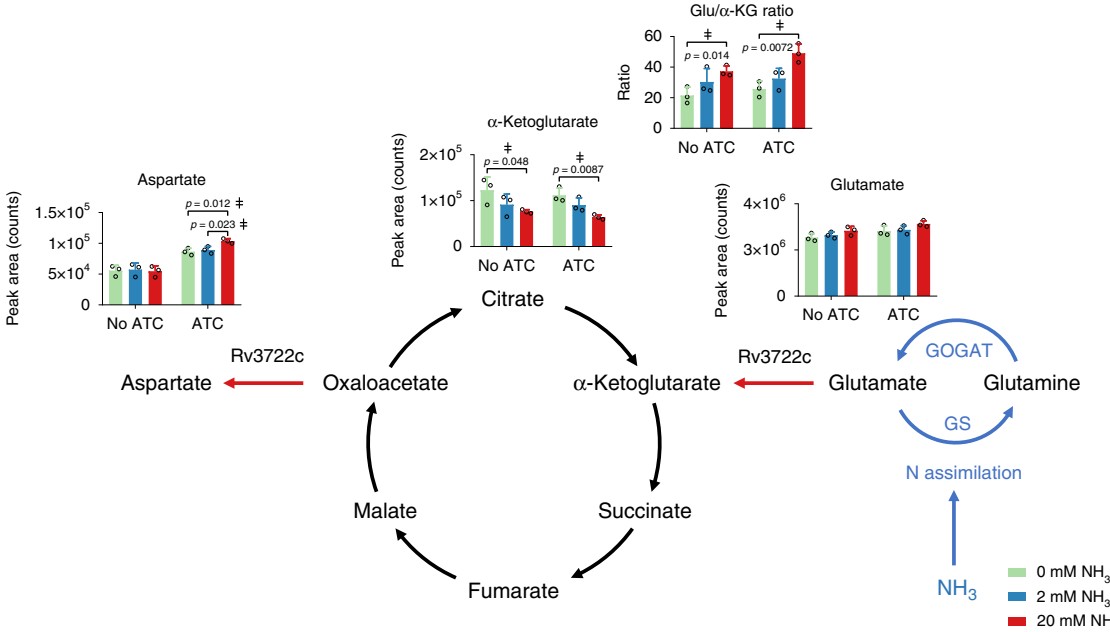

**Fig. 6 Rv3722c links nitrogen assimilation to aspartate synthesis.** Rv3722c-TetON was cultured in modified minimal TSM media supplemented with 2 mM ammonium chloride and 1% casein hydrolysate, with or without ATC, to predeplete Rv3722c. After adaptation (24 h) to the same media without casein hydrolysate and 2 mM ammonium chloride as sole nitrogen source, the media was replaced with the same media containing 0 (green), 2 (blue), or 20 mM (red) ammonium chloride. After 4 h at 37 °C, the relative levels of the indicated metabolites were determined using LC-MS. Blue arrows indicate reactions involved in nitrogen assimilation (GS glutamine synthetase, GOGAT glutamine oxoglutarate aminotransferase), while red arrows indicate the direction of the Rv3722c-mediated reaction. Data are presented as individual datapoints (open circles) and mean +/− SD of three experimental replicates ($n = 3$), representative of two independent experiments. Statistically significant differences were identified by a combination of a two-sided unpaired Student's $t$ test ($p < 0.05$, as indicated) and Mann–Whitney $U$ rank testing (for which the smallest possible $p$-value with $n = 3$ is 0.1) (double dagger); The Glu/αKG ratio was calculated using Excel. Source data are provided as a Source Data file.

we traced the metabolic fate of the alpha amino nitrogen of Glu and Asp in Rv3722c-sufficient and -deficient *Mtb* incubated with α-15N-Asp or α-15N-Glu after 24 h. As expected, we observed 15N labeling of known Asp-dependent metabolites following incubation with α-15N-Asp (Fig. 8a). Incubation with α-15N-Glu resulted in a similar, but Rv3722c-dependent, pattern of 15N labeling (Fig. 8a), demonstrating that Rv3722c plays a nonredundant role in the metabolic distribution of assimilated nitrogen.

Among Asp-dependent metabolic intermediates, we noted that levels of the purine nucleotide IMP were particularly elevated in Rv3722c-deficient *Mtb* with corresponding reductions in the levels of more downstream adenylate nucleotides, both changes being responsive to exogenous aspartate (Figs. 7c and 8b). Moreover, we noted growth of Rv3722c-deficient *Mtb* supplemented with Asp could be further augmented with the addition of hypoxanthine but not adenine. Hypoxanthine is a purine salvage pathway intermediate that bypasses one of the two Asp-dependent steps in adenylate nucleotide biosynthesis (Fig. 8b). Metabolic tracing similarly revealed a selective incorporation of 15N, but not 13C, into adenylate nucleotides in Rv3722c-sufficient-, but not -deficient, *Mtb* (Fig. 8b). These results identify a metabolically essential and specific role for Asp-derived nitrogen, unlinked to its carbon backbone, in *Mtb* growth.

## Discussion

Despite the advent of high-throughput sequencing technologies, access to the biological information encoded within most genomes remains heavily dependent on sequence homology-based inference. While powerful, this approach has proven least effective for genes encoding the potentially most specific or unique biological functions of a given genome. Such approaches

introduce an often unrecognized bias toward annotation of widely conserved, rather than species-specific functions[1]. Here, we applied ABMP to annotate the function of one essential and previously unannotated gene and two previously misannotated genes predicted to encode the same function. Our work highlights that the protein annotation gap is bigger than the number of unannotated proteins and includes a generally underrecognized degree of misannotation.

Our study highlights two types of misannotation, one in which two genes were erroneously annotated as AspATs on the basis of a moderate degree of sequence homology, and another in which the identity of the actual AspAT was unrecognized. In retrospect, the misannotation of Rv0337c/AspC and Rv3565/AspB as AspATs can be explained by structural studies demonstrating that the amino acid sequence of aminotransferases is dominated by their overall fold, rather than active site-specific, architecture[35,36]. Accordingly, while the *Mtb* genome is bio-informatically predicted to encode several aminotransferases[37], only a handful have been biochemically characterized[38–40].

The structure of Rv3722c, in contrast, differs from canonical AspATs, and belongs to a recently proposed new subgroup of AspATs[25]. Members of this subgroup all contain an uncommon auxiliary N-terminal domain and belong to the relatively uncharacterized protein family PF12897, members of which are almost exclusively present in bacteria[7,25]. Though the function of the auxiliary N-terminal domain is unclear, its restricted species distribution and absence in humans represent an opportunity for selective targeting of an enzyme class for which few selective inhibitors currently exist[29,41].

Through a combination of in vitro biochemistry, in vivo metabolomics, and culture experiments, we now unambiguously resolve Rv3722c as the primary AspAT of *Mtb*, Rv0337c as an

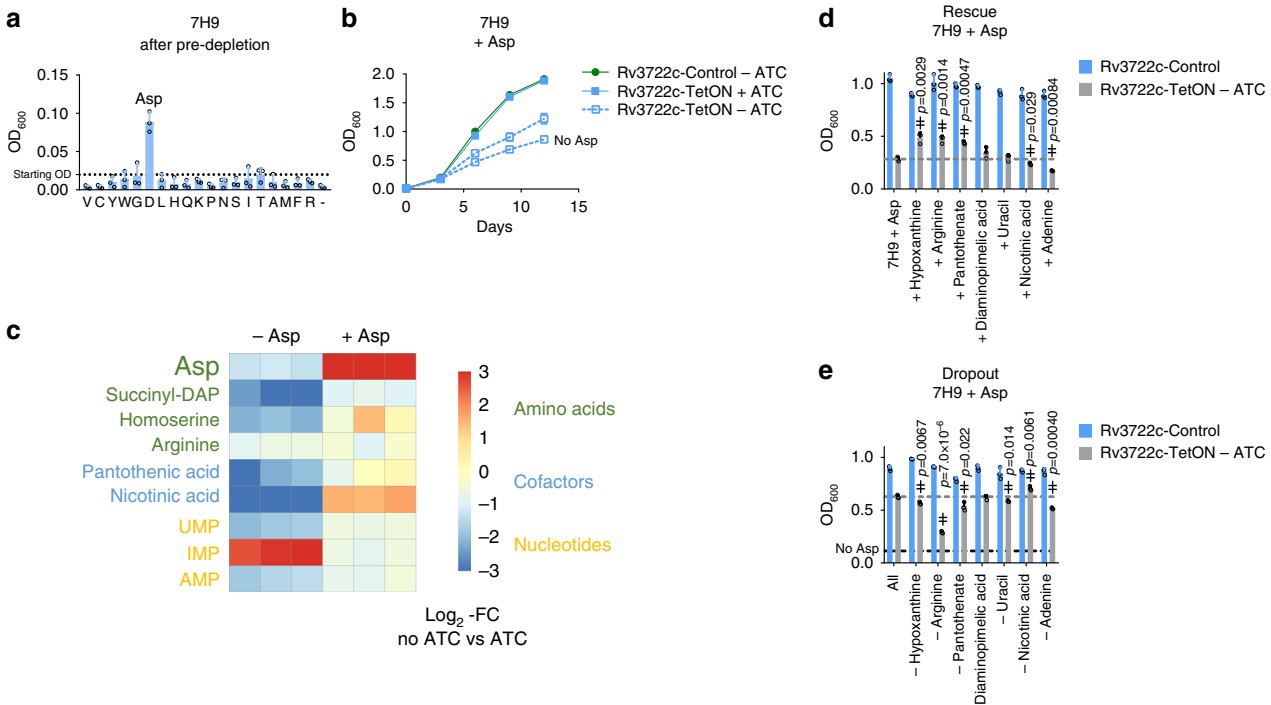

**Fig. 7 Aspartate supplementation rescues growth and metabolic defects. a** Rescue screen with single amino acids. Rv3722c-TetON was cultured in 7H9 with casein hydrolysate without ATC to predeplete Rv3722c. After predepletion, bacteria were transfered to 7H9 supplemented with 3 mM of the indicated amino acids at a final optical density at 600 nm ($OD_{600}$) of 0.02. Bacterial growth was measured after 12 days at 37 °C. Amino acids are indicated by their 1-letter code; 7H9 control. **b** Growth curve of Rv3722c-proficient and -deficient *Mtb* in 7H9 supplemented with Asp. Rv3722c-TetON and Rv3722c-control were cultured in 7H9 supplemented with 3 mM Asp, $+/-$ anhydrotetracycline (ATC). Data from Rv3722c-TetON cultured in the absence of ATC and Asp (Fig. 1a) were plotted as a reference. **c** Heatmap showing levels of Asp and its essential downstream metabolites. Rv3722c-TetON was cultured in 7H9 (Fig. 1a) or 7H9 supplemented with Asp (**b**) with or without ATC for 12 days. Relative levels of indicated metabolites were determined using LC-MS. Colors indicate $log_2$-fold change of the LC-MS peak areas detected without ATC versus those with ATC, in the absence ($-$) or presence ($+$) of 3 mM Asp. **d** Rescue screen with essential Asp metabolites. As **a**, but supplementing with diaminopimelic acid, arginine, uracil (at 1 mM) or nicotinic acid, pantothenic acid, adenine, or hypoxanthine (at 0.15 mM), in addition to 3 mM Asp. **e** Dropout screen with essential Asp metabolites. As **d**, but supplemented with all but the indicated product. Bacterial growth data in **a**, **b**, **d**, and **e** are presented as mean $+/-$ SD of three experimental replicates, with individual data datapoints shown as open circles (**a**, **d**, **e**). Fold changes are presented as mean of a single ratio ($n = 1$ per square). Data are representative of at least two independent experiments. See also Supplementary Fig. 16. Statistically significant differences between culture media were identified by a combination of a two-sided unpaired Student's *t* test ($p < 0.05$, as indicated) and Mann–Whitney *U* rank testing (for which the smallest possible *p*-value for $n = 3$ is 0.1) (double dagger); source data are provided as a Source Data file.

alanine transaminase and Rv3565 as an alanine–valine transaminase, further analyses of which may improve future sequence-based predictions.

Gene annotations aside, growing evidence has implicated nitrogen metabolism as an essential determinant of *Mtb*'s pathogenicity[16]. Mutants for the asparaginase AnsA, for example, exhibited impaired Asn assimilation, and were attenuated in macrophages and mice[42], while mutants for AnsP1 were unable to import Asp in vitro and were attenuated in mice[15]. Inhibition of glutamine synthetase, a key enzyme in nitrogen assimilation, has similarly been shown to be essential for growth in macrophages and guinea pigs[34]. Together, these studies have established nitrogen assimilation and uptake as essential determinants of *Mtb*'s pathogenicity.

Our discovery of Rv3722c as the primary AspAT of *Mtb* and its essentiality in vitro and in vivo now expands this list to include nitrogen distribution. Aminotransferases are pyridoxal phosphate (PLP)-dependent enzymes that catalyze the reversible transfer of nitrogen from an amino donor to a keto acid amino acceptor. Aminotransferases are, therefore, indispensable for the liberation and distribution of nitrogen from amino acids. The aminotransferases involved in nitrogen distribution and their essentiality in *Mtb*, however, remain largely uncharacterized[43].

In vitro, aminotransferases have been shown to have somewhat overlapping substrate specificities, and in vivo, loss of one aminotransferase can often be overcome by artificial overexpression of another[44]. In *E. coli*, for example, only strains lacking both aminotransferases AspC and TyrB are Asp auxotrophs[45]. Aminotransferases have thus come to be viewed as physiologically promiscuous enzymes that catalyze highly redundant reactions[44].

Our studies indicate that *Mtb* has evolved with Rv3722c functioning as its sole AspAT in Glu-containing environments. Glu is the most prevalent metabolite in many mammalian tissues and bacteria[46–48]. Together with Gln, Glu is the primary portal of nitrogen assimilation in most bacteria[49]. Even though Asp is not a part of nitrogen assimilation, an estimated 27% of all assimilated nitrogen is distributed via Asp for the dedicated biosynthesis of several cofactors, nucleotides, and amino acids[50]. Using Glu-based 7H9 culture medium, we show that Rv3722c transfers nitrogen from Glu to Asp, thus linking nitrogen assimilation to Asp-mediated nitrogen distribution. Synthesis of Asp was required for synthesis of key downstream metabolites and in vitro growth (Figs. 1a and 5a). Rv3722c-deficient *Mtb* specifically showed depletion of these downstream metabolites, most of which could be rescued by Asp supplementation (Fig. 7c). While Asp supplementation in 7H9 did not completely rescue growth

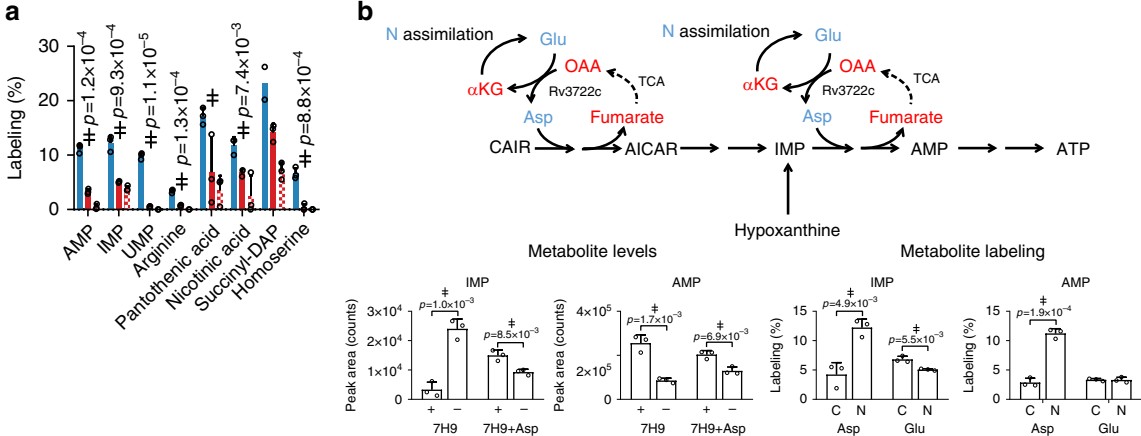

**Fig. 8 Rv3722c governs Asp-dependent nitrogen metabolism. a** $^{15}$N-stable isotope tracing in Rv3722c-proficient *Mtb*. Rv3722c-TetON was cultured in 7H9 supplemented with casein hydrolysate, with and without ATC. After a 3 h adaptation to unsupplemented 7H9, $^{15}$N-Asp (blue) or $^{15}$N-Glu (red) was added to the media at a final concentration of 3 mM. After 24 h at 37 °C, $^{15}$N labeling of the indicated metabolites was determined by LC-MS analysis. **b** Rv3722c serves as a specific nitrogen conduit for de novo purine synthesis. Same as **a**, but using $^{15}$N- and $^{13}$C-labeled Asp and Glu. Nitrogen containing metabolites are depicted in blue, while their carbon backbones are depicted in red. Data are presented as individual points (open circles) and mean +/− SD of three experimental replicates ($n = 3$; but $n = 2$ for succinyl-DAP $^{15}$N-Glu + ATC, due to an analytical interference), representative of two independent replicates. OAA oxaloacetic acid, CAIR carboxyaminoimidazole ribotide, AICAR 5-aminoimidazole-4-carboxamide ribonucleotide. Statistically significant differences between $^{15}$N-Asp + ATC and $^{15}$N-Glu + ATC were identified by a combination of a two-sided unpaired Student's *t* test ($p < 0.05$, as indicated) and Mann–Whitney *U* rank testing (for which the smallest possible *p*-value for $n = 3$ is 0.1) (double dagger); Source data are provided as a Source Data file.

(Fig. 7b), the ability of more downstream Asp-dependent products to augment growth beyond that achieved with Asp alone is a likely indication of a limited flux of exogenous Asp through downstream metabolic pathways. Our results nonetheless demonstrate a nonredundant and essential role for the AspAT activity of Rv3722c.

Previous work suggested that *Mtb* has access to Asp in host macrophages[51] and that Asp serves as an important nitrogen source in vivo. However, mutants deficient in the sole Asp importer Ansp1 were not attenuated in macrophages and only partially attenuated in mice[15]. In line with these results, a recent model of *Mtb* nitrogen metabolism suggests that Asp is not a primary nitrogen source and that most Asp is synthesized inside *Mtb*[52]. The severe attenuation of Rv3722c-deficient *Mtb* in both macrophages and mice confirms that access to Asp is insufficient for *Mtb* to establish an infection in mice and that the main role for Rv3722c lies in both the synthesis and distribution of Asp.

We also found that Rv3722c produces Asp at a rate that is governed by the αKG-Glu ratio, which is an index for the carbon–nitrogen status. It has recently been found that *Mtb* has an extracellular Asp sensor (GlnH) that regulates the carbon–nitrogen status via GlnX, PknG, and finally GarA[53,54]. Moreover, *Mycobacterium smegmatis* strains deficient for PknG or GarA show markedly altered Asp levels[54]. Taken together, these results support a model in which Rv3722c produces Asp at a rate that is co-regulated by the presence of extracellular Asp through the αKG-Glu ratio. Moreover, our results suggest a pivotal role for Asp biosynthesis, in addition to uptake, for full virulence of *Mtb*.

Through the annotation of Rv3722c as the main AspAT in *Mtb* and the link between nitrogen assimilation and Asp biosynthesis on the one hand, and the characterization of the multifaceted essentiality of Asp on the other, we have established Asp biosynthesis as a new potential drug target in *Mtb*.

## Methods

**Chemicals**. All chemicals were from Sigma, unless stated otherwise. All amino acids were in the L-form, unless stated otherwise.

**Strains**. To create an *Mtb* strain with reduced levels of Rv3722c, we employed a protein degradation system previously described[17]. Briefly, a DAS + 4 tag was recombineered into the chromosome of *Mtb* H37Rv, at the 3′-end of the target gene (H37RvMA::Rv3722c-FLAG-DAS or Rv3722c-Control). Next, the FLAG-DAS-tagged mutant was transformed with a StrepR plasmid containing *sspB* downstream of an inducible promoter (H37RvMA::Rv3722c-FLAG-DAS::sspB-pTetON-6 or Rv3722c-TetON)[17]. The *sspB*-expressing plasmid was integrated into the chromosome at the Giles phage integration site. When induced, SspB delivers DAS-tagged protein to the native protease ClpXP for degradation. Regulation was achieved by repression of the *sspB* promoter with a reverse tetracycline repressor (revTetR). RevTetR requires anhydrotetracycline (ATC), which acts as a corepressor, to shut down transcription of *sspB*. Repression of *sspB* suppresses degradation of the DAS-tagged protein. Phenotypically, we thus refer to these mutants as TetON mutants. The recombineering cassette consists of 500 bp flanking sequences around the stop codon of the gene, the DAS tag (inserted at the 3′-end of the target gene), a loxP site, a unique nucleotide sequence ("molecular barcode", to enable pooled analysis of multiple strains if desired), and a hygR selectable marker. The cassette was synthesized as dsDNA (GenScript, Piscataway, NJ) in plasmid pUC57 with flanking *PmeI* sites. The fragment was excised from the plasmid with *PmeI*, and used as a double-stranded DNA recombineering substrate[55]. Strains were confirmed with PCR and drug-resistance phenotypic screening. Rv3722c-TetON was grown in the presence of 500 ng mL$^{-1}$ ATC, which was replenished every 3 to 4 days. Cultures were maintained with 20 μg mL$^{-1}$ streptomycin, but streptomycin was not added during experiments. Bacterial strains generated as a part of this study are available upon reasonable request.

**Culture conditions**. *Mtb* strains were cultured at 37 °C in a biosafety-level 3 facility in 10-mL standing culture flasks containing 7H9 (Middlebrook 7H9 Broth; BD) with glycerol (0.2%), sodium chloride (0.85 g L$^{-1}$), D-glucose (2 g L$^{-1}$), albumin (5 g L$^{-1}$; fraction V, fatty acid-free, Roche) and tyloxapol (0.04%). For specific experiments, strain were grown in 7H9 with added L-amino acids (3 mM) or casamino acids (10 g L$^{-1}$; Hy-Case SF), or in Sauton's (L-Asn, 4 g L$^{-1}$; citric acid, 2 g L$^{-1}$; ferric ammonium citrate, 0.05 g L$^{-1}$; magnesium sulfate, 0.5 g L$^{-1}$; zinc sulfate 1 mg L$^{-1}$; monopotassium phosphate, 0.5 g L$^{-1}$; glycerol, 6%; tween 80, 0.05%; sodium hydroxide ad pH 7).

**Spot assay**. Log-phase *Mtb* cultures were diluted to an OD$_{600}$ of 0.1 in 7H9, followed by serial tenfold dilutions. A volume of 5 μL was spotted onto square plates containing Middlebrook 7H10 agar with or without casamino acids (10 g L$^{-1}$; Hy-Case SF), and with or without 500 ng mL$^{-1}$ ATC. Pictures were taken using a Samsung WB2100 digital camera.

**Rescue with amino acids and downstream products**. Rv3722c-TetON or Rv3722c-Control was precultured in 7H9 with casamino acids, without ATC, to predeplete Rv3722c. After washing with 7H9 without ATC, bacteria were transferred to 96-well plates containing 150 μL culture medium at a final OD$_{600}$ of 0.02.

For rescue with amino acids, medium was supplemented with 3 mM of different amino acids. For rescue with single downstream products, bacteria were transferred to media containing 3 mM Asp, and homoserine, racemic diaminopimelic acid, arginine, uracil at 1 mM, or nicotinamide, pantothenic acid, adenine, and hypoxanthine at 0.15 mM. The same concentration were used for the dropout screen. Bacterial growth was monitored at 600 nm using a platereader (SpectraMax M2e, Molecular devices).

**ABMP**. Activity-based metabolite profiling was performed by incubating a mycobacterial metabolite extract with 10 μM purified protein in 20 mM TRIS-HCl pH 7.4 or a control (heat-inactivated protein or buffer)[18]. At several timepoints, samples (20 μL) were collected into 100 μL ice-cold acetonitrile:methanol:water (2:2:1; v:v:v). After centrifugation (10 min, 21,000 g, 4 °C), the samples were analyzed as described under metabolomics. When described, [15]N-labeled amino acids (10 mM; Cambridge Isotope Laboratories), or keto acids (20 mM) were added to force the reaction in one direction. Untargeted comparisons of metabolite extracts before and after incubation were performed using the XCMS online platform, after converting the raw files to the MzXML format using Proteowizard[56,57].

**Metabolomics**. Metabolomics was performed on liquid cultures grown in media with 0.04% tyloxapol, normalized by OD₆₀₀, after washing twice with ice-cold PBS (5-10 min at 3,000 g, and 4 °C). Washed cells were resuspended in 1 mL ice-cold acetonitrile:methanol:water (2:2:1; v:v:v), bead-beaten (6 times for 30 s at 6500 rpm, using a Precellys 24 with Cryolys cooling unit (Bertin Technologies)) with 0.1 mm Zirconia/silica beads (BioSpec), followed by clarification (10 min, 21,000 g, 4 °C) and filter-sterilization through a 0.22-μm Spin-X filter (Sigma)[18]. Samples were stored at −80 °C until LC-MS analysis. Aqueous normal phase LC-MS metabolomics was performed on 2 μL non-diluted samples, using a Diamond Hydride Type C column (Cogent) on an Agilent 1200 LC system coupled to an Agilent Accurate Mass 6220 Time of Flight (TOF) spectrometer (m/z 50-1200) operating in the positive (amino acids) and negative (keto acids) ionization mode, using a 0.4 mL min⁻¹ gradient of water with 0.2% formic acid (A) in acetonitrile with 0.2% formic acid (B) (0–2 min: 85% B, 3–5 min: 80% B, 6–7 min: 75% B, 8-9 min: 70% B, 10–11 min: 50% B, 11–14: 20% B, 14–24: 5% B, followed by 10 min re-equilibration at 85% B)[18].

For the detection of phosphorylated metabolites, malate, citrate, and succinate, we used an ion-pairing LC-MS method. Samples were injected (5 μL) onto a ZORBAX RRHD Extend-C18 column (2.1 × 150 mm, 1.8 μm; Agilent) with ZORBAX SB-C8 (2.1 mm × 30 mm, 3.5 μm; Agilent) precolumn heated to 40 °C, and separated using a gradient of 5 mM tributylamine/5.5 mM acetate in water: methanol (97:3; v:v)(mobile phase A) and 5 mM tributylamine/5.5 mM acetate in methanol (mobile phase B) at 0.25 mL min⁻¹, as follows: 0–3.5 min: 0% B, 4-7.5 min: 30% B, 8–15 min: 35% B, 20–24 min: 99% B, 24.5–25 min: 0% B; followed by 5 min of re-equilibration at 0% B (Agilent 1290 Infinity LC system). Post-column, 10% dimethylsulfoximide in acetone (0.2 mL min⁻¹) was mixed with the mobile phases to increase sensitivity. Data were collected from m/z 50-1100, using an Agilent Accurate Mass 6230 Time of Flight (TOF) spectrometer with Agilent Jet Stream electrospray ionization source operating in the negative ionization mode (gas temp: 325 °C, drying gas: 8 L min⁻¹, nebulizer: 45 psig, sheath gas temp: 400 °C, sheath gas: 12 L min⁻¹, Vcap: 4000 V, Fragmentor: 125 V). Metabolites were identified based on accurate mass-retention time identifiers for masses exhibiting the expected distribution of accompanying isotopes. The abundance of extracted metabolite ion intensities was determined using Profinder 8.0 and Qualitative Analysis 7.0 (Agilent Technologies). Heatmaps were generated in R (version 3.5.2) using the pheatmap package (version 1.0.12).

**Stable isotope tracing**. *Mtb* strains were cultured in 7H9 with casamino acids with or without ATC to predeplete Rv3722c. To accommodate large volumes, bacteria were grown in closed 1-L flasks in a shaking incubator. At the day of the experiment, the cultures were resuspended in 7H9 without tyloxapol and casamino acids at an OD₆₀₀ of 1. After 3 h, 3 mM [15]N-Glu, [15]N-Asp, [13]C₅-Glu, and [13]C₄-Asp (Cambridge Isotope Laboratories) were added. After 0, 2, 4, 8, and 24 h, samples (10 mL) were collected on ice, spun down, and washed once with cold PBS. Metabolite extraction and LC-MS analysis were performed as described under metabolomics, but using an Agilent Accurate Mass 6545 Quadrupole Time of Flight (Q-TOF). The percentage of labeling was determined using the batch iso-topologue extraction function in Profinder software (Agilent Technologies), which corrects for natural isotope abundance. In some cases, succinyl-DAP co-eluted with and interfering signal at the M + 2 m/z value. For these cases, the sample was excluded from the presented data.

**Nitrogen shift**. Rv3722c-TetON was cultured in modified TSM media[58], containing magnesium sulfate (0.5 g L⁻¹), calcium chloride (0.5 mg L⁻¹), zinc sulfate (0.1 mg L⁻¹), copper sulfate (0.1 mg L⁻¹), ferric iron chloride (50 mg L⁻¹), monopotassium phosphate (0.5 g L⁻¹), dipotassium phosphate (1.5 g L⁻¹), sodium chloride (0.85 g L⁻¹), D-glucose (2 g L⁻¹), albumin (5 g L⁻¹; fraction V, fatty acid-free, Roche), and tyloxapol (0.04%). Bacteria were first grown in modified TSM supplemented with casamino acids (10 g L⁻¹; Hy-Case SF) and 2 mM ammonium chloride, in the presence or absence of ATC to predeplete Rv3722c. To

accommodate large volumes, bacteria were grown in closed 1-L flasks in a shaking incubator. Next, the bacteria were transferred to the same media without casamino acids and tyloxapol, and incubated at 37 °C. After 24 h, the media was replaced with modified TSM containing 2 mM ammonium chloride (1X NH₃), 20 mM ammonium chloride (10× NH₃), or no ammonium chloride (0× NH₃). Samples (10 mL) were collected, followed by centrifugation (10 min, 3000 g, 4 °C). The pellet was resuspended in ice-cold acetonitrile:methanol:water (2:2:1; v:v:v), and processed and analyzed as described under metabolomics, but using an Agilent Accurate Mass 6545 Quadrupole Time of Flight (Q-TOF).

**Mouse infection**. Prior to infection, Rv3722c-TetON was grown in Sauton's media with or without 500 ng mL⁻¹ ATC for 10 days to predeplete Rv3722c. Six- to 8-week-old female BL/6 mice (Jackson Laboratory, Bar Harbor, ME) were infected in an aerosol chamber Madison chamber (University of Wisconsin). Immediately prior to infection, cells were sonicated and diluted 1:100 in PBS, and ~100–300 CFUs administered to each mouse. Mice were fed regular chow, or chow containing doxycycline at 2000 ppm (Research Diets, New Brunswick, NJ). Lungs were harvested and plated for CFUs on Middlebrook 7H10 agar or Middlebrook 7H9 + 1.5% Bacto Agar (Difco) containing OADC enrichment (Middlebrook) and 0.2% glycerol with ATC. Three to five mice per group were harvested at the indicated timepoints. To determine the number of non-ATC-regulatable mutants, two lungs per group collected after 2, 3, and 6 weeks were plated onto plates with and without ATC. In a similar experiment, mice were infected with pre-depleted Rv3722c-TetON and wild-type, and fed chow without ATC. Lungs from wild-type mice and Rv3722c-TetON were plated on plates without ATC, or plates with and without ATC, respectively. Mice were housed at 19–23 °C, a humidity of 35–60%, and a 14 h/10 h light/dark cycle. The animal study protocol for mouse infections was approved by the Harvard Medical Area Institutional Animal Care and Use Committee. We have complied with all relevant ethical regulations pertaining to these animal experiments.

**Macrophage infection**. Femoral mouse bone marrow cells were isolated from 7 to 8-week-old male C57BL/6 J mice (Jackson Laboratory) and cultured (37 °C, 5% CO₂) on Petri dishes containing DMEM with 1% HEPES (Gibco), 10% FBS, and 20% L929 cell-conditioned medium (LCM) as a source of macrophage colony-stimulating factor[59]. On day 6, the cells were seeded in 96-well plates containing the same medium with 10% LCM at a density of 6 × 10⁴ cells per well. When indicated, cells were activated by adding 50 ng mL⁻¹ recombinant mouse IFN gamma (Invitrogen). On day 7, the cells were infected with bacteria (cultures in 7H9 with casamino acids in the presence or absence of 500 ng mL⁻¹ ATC to predeplete Rv3722c) in triplicate (multiplicity of infection of 1) for 4 h, followed by two washes with PBS and the addition of fresh medium with or without 500 ng mL⁻¹ ATC. On day 0, 3, and 5 post infection, the macrophages were lysed in 0.01% Triton X-100 and plated onto 7H10 plates containing 10 g L⁻¹ casamino acids and 500 ng mL⁻¹ ATC at serial dilutions for determination of CFUs. The animal study protocol for harvesting of mouse femoral bone marrow cells was approved by the Institutional Animal Care and Use Committee of Weill Cornell Medicine. We have complied with all relevant ethical regulations pertaining to these animal experiments.

**Western blotting**. Cells cultured in 7H9, 7H9 with casamino acids or Sauton's were spun down, resuspended in 1 mL ice-cold lysis buffer (100 mM NaCl, 5% glycerol, 1 mM dithiothreitol, and protease inhibitor cocktail (cOmplete, EDTA-free; Roche) in 50 mM Tris-HCl, pH 8), and transferred to bead-beating tubes containing 0.1 mm Zirconia/silica beads (BioSpec). After bead-beating three times for 30 s at 6500 rpm in a cooled bead-beater (Precellys tissue homogenizer), samples were clarified and filter-sterilized as described under metabolomics. Samples were mixed with Laemmli sample buffer (Bio-Rad) with β-mercaptoethanol, and denatured (10 min, 95 °C). Approximately 7.5 μg of the total protein (as determined by A280 on a Nanodrop; Thermo Fisher Scientific) was loaded onto 8–16% TGX gels (Bio-Rad) and run in SDS-PAGE buffer for 30 min at 90 V followed by 45 min at 150 V. Proteins were transferred onto Protran 0.2-μm nitrocellulose blotting membranes (Amersham) (300 mA, 100 min, 4 °C; or 30 mA overnight, 4 °C). After washing (PBS-Tween, 30 min, room temperature) and blocking (Odyssey blocking solution; LI-COR Biosciences, 1 h, room temperature), the membranes were incubated with monoclonal mouse anti-FLAG antibody M2 (Sigma; 1:400) and rabbit anti-prcB (a gift from G. Lin and C. Nathan, 1:10,000) overnight at 4 °C. Next, the membranes were incubated with IRDye 800CW goat anti-mouse IgG (LI-COR Biosciences, 1:20,000) and IRDye® 680LT Donkey anti-Rabbit IgG (H + L) (LI-COR Biosciences, 1:20,000) secondary antibodies for 2 h at room temperature in Odyssey blocking buffer:PBS-Tween (1:1). After washing, proteins were detected with the Odyssey Infrared Imaging System using the 700 and 800 nm channels at a resolution of 169 μm (LI-COR Biosciences). The brightness and contrast of the obtained full gel images were optimized for both channels, using Image Studio Lite (LI-COR Biosciences). If required, the amount of loaded protein was normalized based on the intensity of the proteasome subunit B (PrcB) loading control band in exploratory Western blots.

**Protein expression and purification.** For ABMP, *Rv3722c* was cloned in a pET23 (+) vector encoding a C-terminal His₆-tag, and transformed into Bl21-AI cells (Thermo Fisher). Expression was induced at 37 °C, with 0.2 mM IPTG and 0.2% L-arabinose for 18 h. Cells were spun down, frozen at −80 °C, resuspended in 2× PBS with 10 mM imidazole, 1 mM pyridoxal phosphate, 5% glycerol, protease inhibitors (cOmplete, EDTA-free; Roche) and 1 μg mL⁻¹ DNAse I, and disrupted using an Emulsiflex C5 (Avestin). After centrifugation, the lysate was loaded onto a Ni²⁺ column (HiPrep IMAC FF 16/10; GE Healthcare), followed by elution with a gradient of 10–250 mM imidazole in 2× PBS with 10 mM imidazole, 1 mM pyridoxal phosphate, 5% glycerol. Rv3722c-containing fractions were pooled and concentrated over Amicon ultra 30 KDa cutoff filter (Millipore). After buffer exchange with 50 mM Tris-HCl pH 8 with 5% glycerol, the protein was loaded onto a HiPrep Q HP 16/10 and eluted with a gradient of 0–1 M NaCl in 50 mM Tris-HCl pH 8 with 5% glycerol. Rv3722c-containing fractions were concentrated and the buffer was exchanged with 50 mM Tris-HCl pH 8 with 5% glycerol and 100 mM NaCl.

For structure determination, Rv3722c cloned into a modified pET28 vector encoding a TEV protease cleavage site followed by a C-terminal His₆-tag was chemically transformed into *E. coli* BL21 (DE3) cells. Protein production and purification were performed as outlined above, albeit with minor modifications. Here, 0.1 mM IPTG was used to induce overnight expression at 18 °C. In addition, after Ni-NTA affinity purification, the His₆-tag was removed using TEV protease, and the protein was subjected to size-exclusion chromatography (HiPrep 26/60 Sephacryl S-200; GE Healthcare) using a buffer consisting of 40 mM HEPES pH 7.4 and 150 mM NaCl. Fractions of pure Rv3722c were pooled, concentrated, flash frozen and stored at −80 °C until further use.

For ABMP, Rv0337 and Rv3565 were produced and purified using a similar approach. However, the overexpression and production of both enzymes was carried out in *E. coli* C41 (DE3) cells. Furthermore, before Rv0337 was eluted off the Ni-NTA column, an additional step, adapted and modified from ref. [60], was introduced to remove contaminating chaperones.

**Ketoglutaramate preparation.** Ketoglutaramate was prepared and purified based on the method described by Jaisson et al.[61]. Glutamine (146 mg) was oxidized by *Crotalus adamanteus* L-amino acid oxidase (10 mg, Sigma) in the presence of catalase (10 units, Sigma) for 20 h in a shaking incubator (37 °C, 450 rpm). After protein precipitation with perchloric acid, the reaction mixture was purified over an AG 50W-X8 column (Bio-Rad), neutralized using potassium carbonate, and dried under a vacuum. Further purification was performed on a strong anion exchange solid phase extraction cartridge (Bond Elut SAX, 3 mL, Agilent) by loading 3 mL of the acidified (0.2% formic acid in water) crude ketoglutaramate and collecting the flow-through. The MS/MS fragmentation spectra were obtained using the LC-MS method described under metabolomics, but using an Agilent accurate mass 6545 quadrupole-time of flight (Q-TOF) spectrometer operated at a collision energy of 10, 20, and 40 V.

**Enzyme activity assays.** Substrate screens for Rv3722c were performed by incubating 1 μM protein, 1 mM amino acid, and 10 mM keto acid (αKG, oxaloacetate or pyruvate) in 150 μL 10 mM Tris-HCl pH 7.4 containing 10 μM pyridoxal phosphate at 37 °C in triplicate. Samples were collected at 0, 5, 10, 30, and 60 min, and quenched in ice-cold acetonitrile:methanol (1:1) spiked with 0.1 mM ¹³C₅,¹⁵N-L-glutamic acid, ¹³C₄,¹⁵N-L-aspartic acid, or ¹³C₁-L-alanine (Cambridge Isotope Laboratories) as internal standard for reactions with αKG, oxaloacetate or pyruvate, respectively. Samples were stored at −80 °C until RapidFire analysis.

Rv3722c enzyme kinetics were determined by incubating purified protein (0.01 μM for L-Asp and L-Kyn, 1 μM for other substrates) with 10 mM αKG and amino acid concentrations ranging from 0.05 to 20 mM in 10 mM Tris-HCl pH 7.4 containing 10 μM pyridoxal phosphate at 37 °C in triplicate. Samples were collected at 0, 0.5, 1, 2, 5, 10, and 20 min, and quenched in ice-cold acetonitrile:methanol (1:1) spiked with 0.1 mM ¹³C₅,¹⁵N-L-glutamic acid as internal standard (Cambridge Isotope Laboratories). Samples were stored at −80 °C until RapidFire analysis. Measured initial velocities were fitted to Michaelis–Menten kinetics using Graphpad Prism 7 software.

Ninety-six-well plates containing enzyme activity samples were analyzed using a RapidFire high throughput MS system coupled to a 6495 triple quadrupole mass spectrometer (Agilent). Samples were loaded onto a HILIC type H1 cartridge with acetonitrile with 0.1% formic acid, followed by a wash with the same mobile phase. Elution was performed with 20% acetonitrile in water with 0.1% formic acid. The settings were as follows: Aspirate: 600 ms; Load/Wash: 100 ms; Elute: 5000 ms; Re-equilibrate: 4000 ms; flow: 1.25 mL min⁻¹. The mass spectrometer was set to trace the following transitions in positive ionization mode: Glutamate: m/z 148−>84, CE 16 V; ¹³C₅,¹⁵N-Glutamate: m/z 154−>89, CE 16 V; Aspartate: m/z 134−>74, CE 10 V; ¹³C₄,¹⁵N-Aspartate: m/z 139−>77, CE 10 V; Alanine: m/z 90−> 44, CE 15 V; ¹³C-Alanine: m/z 91−> 45, CE 15 V.

Rv0337c/AspC and Rv3565/AspB enzyme kinetics were determined by a coupled reaction with lactate dehydrogenase. Purified recombinant enzymes (1 μM) were incubated with 10 mM keto acid (sodium 3-methyl-2-oxobutyrate (keto-Val), (±)-3-methyl-2-oxovaleric acid sodium salt (keto-Ile), sodium 4-methyl-2-oxovalerate (keto-Leu), 2-oxoadipic acid (keto-aminoadipic acid), α-keto-γ-(methylthio)butyric acid sodium salt (keto-Met) and αKG), and L-Ala

concentrations ranging from 0 to 20 mM, in 100 mM Tris-HCl pH 7.4 containing 10 μM pyridoxal phosphate, 0.5 U mL⁻¹ L-lactate dehydrogenase from rabbit muscle (Roche) and 1 mM NADH (Roche) at 37 °C in triplicate. The disappearance of NADH was followed spectrophotometrically at 340 nm using a SpectraMax M2e (Molecular Devices) microplate reader. Measured velocities were fitted to Michaelis–Menten kinetics using Graphpad Prism 7 software.

**Protein crystallization and structure determination.** Rv3722c (~40 mg mL⁻¹) was co-crystallized with ligands by sitting drop vapor diffusion at 17 °C. Rv3722c pre-incubated with 5 mM L- glutamic acid at 25 °C for an hour was added to an equal volume of reservoir solution containing 100 mM sodium acetate pH 4.5, 200 mM Li₂SO₄, and 50% PEG 400. Similarly, Rv3722c pre-incubated with 10 mM L-kynurenine was mixed with a reservoir solution containing 100 mM Na₂HPO₄: citric acid pH 4.2, 40% ethanol and 5% PEG 1000. Before data collection, the crystals of Rv3722c/L-Kynurenine complex were cryoprotected in the mother liquor containing 25% glycerol and flash frozen in liquid nitrogen. Cryoprotection was not necessary for crystals of Rv3722c/L-glutamic acid complex.

Diffraction datasets for Rv3722c/L-glutamic acid were collected on beamline 23 ID at Argonne National Laboratory APS synchrotron. The data were indexed, integrated, and scaled using PROTEUM3 software (Version 2016.2, Bruker AXS Inc). Data were truncated in CCP4 suite[62], and the structure was solved by molecular replacement using the unliganded structure of the same enzyme (PDB 5C6U [https://www.rcsb.org/structure/5C6U]) as a search model in MOLREP[63].

Data for Rv3722c/L-kynurenine were also collected on beamline 23 ID at Argonne National Laboratory APS synchrotron. Data were auto processed in XDS[64]. The unmerged data were corrected for anisotropy using the STARANISO webserver [http://staraniso.globalphasing.org/cgi-bin/staraniso.cgi]. The structure was solved by molecular replacement as described above.

The models were iteratively refined in PHENIX[65] and built manually in COOT[66]. Ligand models and dictionary files were created in ELBOW BUILDER from the PHENIX[65] suite and fitted into the density in COOT[66]. Ligand OMIT maps were calculated using Polder Maps[67] in PHENIX. All figures were prepared in Chimera[68]. Data collection and refinement statistics are given in Supplementary Table 1. The coordinates and maps for Rv3722/Glu and Rv3722/KYN have been deposited into the Protein Data Bank under accession codes 6U78 [https://www.rcsb.org/structure/6U78] and 6U7A [https://www.rcsb.org/structure/6U7A], respectively.

**Phylogenetic distribution.** A precomputed bacterial distribution of pfam family PF12897 was downloaded from Annotree (AnnoTree v1.1.0; GTDB Bacteria Release 03-RS86; Pfam v27.0; E-value of 0.00001)[69] and visualized in Interactive Tree of Life[70].

**Statistical methods.** We applied a combination of statistical tests to detect statistically meaningful differences for comparisons involving a sample size of three. First, Student's *t* test was applied to test for a significant difference in group means ($p < 0.05$). Given the sensitivity to outliers of Student's *t* test when applied to small sample sizes, we additionally applied the alternative nonparametric Mann–Whitney *U* test to test if the sample groups were also separated by ranks (for which the smallest possible *p*-value with a sample size of three is 0.1).

**Reporting summary.** Further information on research design is available in the Nature Research Reporting Summary linked to this article.

## Data availability
Structural data that support the findings of this study have been deposited in the Protein Data Bank with the accession codes 6U78 (Rv3722/Glu) and 6U7A (Rv3722/KYN). The source data underlying Fig. 1a–c, e, f; 2a–b; 3a, b, d; 5a, b; 6; 7a–c; and 8 a–b; and Supplementary Figs. 1a, b; 2; 3a,b; 4a-c; 5; 11a-e; 12a-c; 13a,b; 14a,b; 15 and 16 are provided as a Source Data file. Metabolomics datasets that support the main findings of this study are available from the Metabolights repository with the accession code MTBLS1559. The data underlying the phylogenetic tree for PF12897 were downloaded from Annotree (AnnoTree v1.1.0; GTDB Bacteria Release 03-RS86; Pfam v27.0; E-value of 0.00001)[69].

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

## Acknowledgements

We thank Carl Nathan for critical reading of the paper, Myung Hee Lee for expert statistical consultation, and the Bill and Melinda Gates Foundation TB Drug Accelerator Program (OPP1177930), NIH Tri-I TBRU (U19-AI11143), NIH NIAID Functional Genomics Program (U19-AI107774), and the Potts Memorial Foundation for support.

## Author contributions

R.J. performed and analyzed all western blots, enzyme kinetics, macrophage infections, in vitro culture, and metabolomics experiments; L.M., R.H., R.J., and B.S. cloned, expressed, and purified aminotransferases; L.M., R.H., and J.S. crystalized proteins and solved their structures; S.W., K.G., and E.R. performed and analyzed the mouse infection experiments; J.P. generated the Rv3722c mutant; R.J., K.R., K.G., E.R., L.M., and J.S. designed the experiments; and R.J., K.R., L.M., and J.S. wrote the paper. All of the authors have read, edited, and approved the paper.

## Competing interests

The authors declare no competing interests.
