## [Peer Review File · Nature Communications]

Reviewers' comments:

Reviewer #1 (Remarks to the Author):

The manuscript by Jansen et al. is an elegant study deciphering the role Rv3722c plays in nitrogen metabolism of *M. tuberculosis*. The manuscript is well written and the experiments/results are presented with a logical flow, establishing Asp biosynthesis as a new potential drug target against Mtb. The most exciting finding of this study to me is the different roles the Asp nitrogen and carbon backbone play in Mtb metabolism. The findings are novel and are most definitely of interest to others in the community and the wider field and, therefore, I support the publication of this manuscript in Nature Communications. I only have a few minor suggestions to improve the manuscript, as outlined below.

Line 79; and CULTURED for 2 weeks ...

Line 80; Growth curve OF ...

Line 138; mention the unit for collusion energy

Line 224; AspC and AspB are annotated as which I subclass?

Line 377; a moderate degree of ...

Line 447; levels of Rv3372c ...

Line 523; heatmap

Lines 530 & 548; 1 L flasks?

Line 557; was grown IN Sauton's ...

Line 581; The protocol for what? was approved ...

Line 785; from C? of ...

Line 786; intermediate.

Line 806; How was the Asp modelling carried out?

Table S1; Rv3372c/Glu

Line 897; have been shown as wires ...

Line 903; to move inwards ...?

Consistent use of CFU's and AspAT's throughout the manuscript

Indicate the N-terminal auxiliary domain in S6 for clarity.

Could the authors comment on the following observations;

- High Km values for Asp (Fig 3) considering that Asp is shown to be the preferred substrate for Rv3372c. Likewise, on the Km values for AspC.
- Any explanations as to why Glu could only be observed in one chain? Would crystal contact play a potential role?

Ghader Bashiri

Reviewer #2 (Remarks to the Author):

The manuscript by Jansen et al. describes very interesting and relevant work into the discovery and characterization of the sole aspartate aminotransferase of *M. tuberculosis* encoded by Rv3722c. The two genes previously annotated as aspartate transaminases (aspB and aspC) turn out to be aminotransferases of valine and alanine. The authors elegantly show with activity-based metabolite profiling and metabolomics, that Rv3722c catalyzes the reversible reaction of AspAT and is an important node in nitrogen metabolism of Mtb. The discovery of the AspAT of Mtb and its essentiality in the mouse model also challenges previous work that proposed uptake of aspartate from the host is an important assimilatory process for Mtb survival. The study by Jansen shows

that Mtb mainly relies on its own biosynthetic process for aspartate in vivo, just like it is the case for so many other essential amino acids. Finally, the manuscript describes possible reasons for why aspartate supplementation is not completely rescuing growth of Rv3722c-TetOn -ATC.

While overall this work describes many important contributions to the field there are several concerns and areas that need improvement, better explanations and deeper probing.

1. Figure 1. It is unclear why strain Rv3722c-TetOn -ATC is still growing in Figure 1A past day 6. It looks like that the protein is completely depleted on day 6 yet growth is still observed.
2. Line 60. It is stressed that the attenuation is seen in GLU-based 7H9 medium. Does this mean that this phenotype is GLU-dependent? What is its phenotype in the absence of GLU?
3. Figure S1 shows that Rv3722c Tet-On grows like WT in Sauton's medium. Does this mean that Asparagine rescues the phenotype or is it the absence of glutamate? This should be worked out.
4. The mouse experiment in figure 2B and its repeat experiment in Figure S2 are confusing. Figure 2B clearly shows no growth of pre-depleted Rv3722-TetOn in mouse lungs. However, Figure S2 shows that Rv3722-deficient MTB (ATC-responsive) are multiplying in the lungs to a burden of 10⁴. Then at later time points this population is taken over by escape "mutants" that were non-responsive to ATC. There are several issues and questions that arise from these two experiments.
 - a. Reproducibility of experiment 2B is not demonstrated. In fact, in the repeat experiment, pre-depleted Rv3722-TetOn can grow to a burden of more than 10000 per lung even in the absence of doxycycline. Then the population is taken over by escape mutants.
 - b. Given these discrepancies the normalization in Figure 2B should be removed and the real numbers shown as this might hide a possible reason for this discrepancy. As stated in the figure legend, the number of Rv3722-TetOn was lower than that of wildtype. A lower burden at 24 hours could explain why escape-mutants do not arise in Figure 2B while in Figure S2 they arise after 3 weeks. However, this still cannot explain why in one experiment pre-depleted Rv3722-TetOn multiplies in the first 2 weeks and in the other it doesn't.
 - c. The nature of the reversion is not revealed in the manuscript. This should be shown and discussed.
 - d. It is unclear what non-ATC responsive means (see figure S2). If Rv3722-TetOn is not ATC responsive anymore, it means it should not grow on plates without ATC. However, the opposite is the case, Rv3722-TetOn even grows on plates without ATC. Maybe ATC-dependent/independent (as used in Figure 2B) is the better choice.
 - e. It is common in the field that not only lung burden but also burden in at least one other organ are shown (usually spleen).
5. All in vitro growth curve experiments 1A, 1B, 1F, S1, 7B, don't seem to have error bars.
6. In Figure 3A it is not entirely clear what is meant by "features related to" the feature with m/z 144.05 means. Do you mean fragments?
7. Based on the biochemical experiments it is clear that Rv3722c is the dedicated aspartate aminotransferase, yet no attempts were made to get the structure with Asp bound?
8. Metabolomics methods section: quenching in cold PBS is not specific enough. What was the temperature of PBS, how were the cells washed (centrifuge speed, temperature and time)?
9. Media preparation: was the medium pH adjusted when 3 mM amino acids were added? Certain amino acids at such high concentrations, change the pH of 7H9 drastically.
10. Is it possible that due to the high glutamate concentration in the 7H9 medium the Rv3722-depleted cells have a much lower aKG level and would need aKG supplementation in addition to aspartate in order to fully complement the growth inhibition? Indeed, looking at Figure S15, addition of Casein shows a significant increase of aKG.
11. Why was Sauton's medium not used to find the growth complementing component? In Sauton's medium Rv3722-depleted cells grow like WT, but it is not clear why? Given the limited amount of media components in Sauton's it would be easier to determine which component rescues Rv3722-depleted cells in this medium.

Reviewer #3 (Remarks to the Author):

The manuscript titled "Rv3722c governs aspartate-dependent nitrogen metabolism in *Mycobacterium tuberculosis*" by Jansen et al., is an effort in the identification of the previously misrecognized/underrecognized role of Rv3722c as an aspartate aminotransferase of *Mycobacterium tuberculosis*. Although the manuscript is reasonably well-written and the results are nicely presented using several high-tech methods including TetON system, metabolomics, and structural biology, there are three major issues.

Major comments;

1. Rv3722c is already annotated as an aminotransferase of Mtb with the X-ray structure possessing an aspartate aminotransferase domain (<https://www.ebi.ac.uk/pdbe/entry/pdb/5C6U>), although the authors have added the multifaceted essentiality of Rv3722c to this manuscript. Thus, the rationale for Rv3722c should be reconsidered in page 3, lines 40-44 in introduction section. In the same context, the authors should describe the reasons more clearly for how and why the authors chose and investigated rv3722c among many unrecognized/unidentified genes in Mtb in introduction section.

2. In Figure 1, Figure 2, Figure S1, and Figure S2, Rv3722c-TetOn seems essential for the Mtb's growth in 7H9 broth but is dispensable in Sauton's medium. In particular, results of in vivo experiments are intriguing. However, the authors should investigate the opposite way in vivo experiments between Figure 2B and Figure S2. For example, the authors proved in vitro essentiality of Rv3722c by comparing the growth of Rv3722c-TetOn with/without ATC between two different media. With these results, the authors should investigate in vivo essentiality of Rv3722c-TetON cultured from 7H9 in mice with ATC (Doxycycline) in Figure 2B. In the same line, the initial infectious doses are quite different between Figure 2B and Figure S2, although the authors displayed Log10 normalized data in Figure 2B. Thus, I suggest that the author perform the experiment of Figure 2B with a high infectious dose (>1,000 initial dose or more) to prove whether this gene is virtually essential in vivo.

3. The authors very nicely described the procedures of metabolomics techniques such as TOF MS and Q-TOF to identify the metabolites in pages 28 - 29 in Materials and Methods. However, it would be highly helpful to the readers if the authors add information how to get ID of metabolites such as retention time and spectrum data for each metabolite as supplementary tables. For example, it is helpful to provide comparison data for retention time (RT) with standard in TOF MS. For another example, please provide transition values to figure out ion fragment pattern and RT in Q-TOF analysis.

Minor comments;

1. In line 135, 2,5 h -> 2.5 h?

2. C8 and Rv3722c are co-transcribed by RT-PCR (Arnvig and Young, 2009). Also, DNA polymerase III subunit dnaZX (Rv3721c) may lie in the same operon? Please discuss the downstream effect of Rv3722c-TetON if those genes are affected.

3. In lines 436-439 in page 25, the authors suggest Asp biosynthesis may be a new potential drug target in Mtb. However, Mtb goes through the latency, so what if Rv3722c and the relevant process facilitate the latency? If so, this may lead to more drug tolerance. For example, Figure 2A, Rv3722c-TetON is more resistant in presence of IFN-gamma inside macrophages because there is no difference in Rv3722c-TetON growth regardless of IFN-gamma. Thus, I am wondering drug susceptibility of the Rv3722c-TetON to INH and RIF.

4. Throughout the manuscript, the authors statistically used unpaired Student's t-test by representing mean-/+ SD. However, the sample size is not sufficient to follow this statistic analysis. Thus, please use Mann-Whitney test as a nonparametric test. In addition, the number of mice is too small to conclude in vivo testing.

5. In Figure 1, please present virtual CFUs rather than O.D. value.
6. In line 86 of page 6, macrophages are not in vivo experiment. I am wondering the initial CFUs in macrophages are same between Figure 2A experiments in absence/presence of IFN-gamma? How dose nitric oxide affect the growth of WT and Rv3722c-TetON in macrophages by treating IFN-gamma? In Figure 2B, 0 is correct even though the authors are expressed the normalized data?
7. In line 461, *PmeI* -> italic
8. In line 573, clarify LCM and L cell.

Reviewers' comments:

Reviewer #1 (Remarks to the Author):

The manuscript by Jansen et al. is an elegant study deciphering the role Rv3722c plays in nitrogen metabolism of M. tuberculosis. The manuscript is well written and the experiments/results are presented with a logical flow, establishing Asp biosynthesis as a new potential drug target against Mtb. The most exciting finding of this study to me is the different roles the Asp nitrogen and carbon backbone play in Mtb metabolism. The findings are novel and are most definitely of interest to others in the community and the wider field and, therefore, I support the publication of this manuscript in Nature Communications. I only have a few minor suggestions to improve the manuscript, as outlined below.

Thank you. We have made the following revisions to the text as suggested by the reviewer below:

Line 79; and CULTURED for 2 weeks ...

We have changed "cultures" to "cultured".

Line 80; Growth curve OF ...

We have removed one "of".

Line 138; mention the unit for collision energy

We added "V" here and throughout the text.

Line 224; AspC and AspB are annotated as which I subclass?

AspB and AspC belong to subclass Ia, which we have now added to the text (line 248-249).

Line 377; a moderate degree of ...

We have removed one "of".

Line 447; levels of Rv3372c ...

We have added "of".

Line 523; heatmap

The name of the R package is actually pheatmap, so we have left the text as it was.

Lines 530 & 548; 1 L flasks?

We have changed "1000" to "1".

Line 557; was grown IN Sauton's ...

We have added "in".

Line 581; The protocol for what? was approved ...

We have now clarified the statement regarding the animal use protocols (line 604-607 and 620-623).

Line 785; from C? of ...

We have changed C? to Ca

Line 786; intermediate.

We have changed "36" into the right reference.

Line 806; How was the Asp modelling carried out?

We apologize for the confusion. The word "model" is a misnomer here. To illustrate similarities between the amino acid Asp and the aliphatic backbone of L-Kyn, we superimposed Asp onto the electron density of the aliphatic chain of L-Kyn. Lines 234-235 and 865-866 pertaining to this comment have been rephrased accordingly to indicate how comparisons between Asp and KYN were performed.

Table S1; Rv3372c/Glu

We have changed Rv3722/Glu to Rv3722c/GLU

Line 897; have been shown as wires ...

We have changed "are shown" to "have been shown".

Line 903; to move inwards ...?

We have corrected the sentence into "to move inwards to interact with the ligand"

Consistent use of CFU's and AspAT's throughout the manuscript

We now consistently use CFUs and AspATs throughout the manuscript.

Indicate the N-terminal auxiliary domain in S6 for clarity.

We have now added a Fig S6A to indicate the N-terminal auxiliary domain and core domain.

Could the authors comment on the following observations:

- High K_m values for Asp (Fig 3) considering that Asp is shown to be the preferred substrate for Rv3372c. Likewise, on the K_m values for AspC.

Although high in comparison to many other enzymes, the K_m values of Rv3722c for Asp, and Rv0337c for Ala, are within the range of reported values for other related AspATs (Nobe, JBC, 1998: 1.7 mM; Yagi, Methods Enzymol, 1985: 1.3 mM; Rakhmanova, Biochemistry, 2006: 0.35 mM and 0.75 mM) and, in the case of AspC, AlaTs (Duff, Arch Biochem Biophys, 2012: ranging from 0.21-5.15 mM for various species of plant yeast and bacteria). We have added these references to the text (line 172-173 and 255). We reason that the seemingly high K_m values for these amino acids reflect their intracellular concentrations (Bennett, Nat Chem Biol, 2009).

- Any explanations as to why Glu could only be observed in one chain? Would crystal contact play a potential role? *There is no straightforward explanation to why we observed Glu in only one chain. Crystal contacts in the asymmetric unit do not appear to play a significant role, as all the active sites of the chains are solvent accessible – thus competent to bind the ligand.*

Reviewer #2 (Remarks to the Author):

The manuscript by Jansen et al. describes very interesting and relevant work into the discovery and characterization of the sole aspartate aminotransferase of *M. tuberculosis* encoded by Rv3722c. The two genes previously annotated as aspartate transaminases (aspB and aspC) turn out to be aminotransferases of valine and alanine. The authors elegantly show with activity-based metabolite profiling and metabolomics, that Rv3722c catalyzes the reversible reaction of AspAT and is an important node in nitrogen metabolism of *Mtb*. The discovery of the AspAT of *Mtb* and its essentiality in the mouse model also challenges previous work that proposed uptake of aspartate from the host is an important assimilatory process for *Mtb* survival. The study by Jansen shows that *Mtb* mainly relies on its own biosynthetic process for aspartate in vivo, just like it is the case for so many other essential amino acids. Finally, the manuscript describes possible reasons for why aspartate supplementation is not completely rescuing growth of Rv3722c-TetOn -ATC.

While overall this work describes many important contributions to the field there are several concerns and areas that need improvement, better explanations and deeper probing.

1. Figure 1. It is unclear why strain Rv3722c-TetOn -ATC is still growing in Figure 1A past day 6. It looks like that the protein is completely depleted on day 6 yet growth is still observed.

*We apologize for the confusion. We believe this apparent discordance reflects the limited sensitivity of the anti-FLAG antibody relative to the catalytic activity of Rv3722c, such that the continued growth of Rv3722c past day 6 following removal of ATC likely reflects low/residual levels of Rv3722c-encoded catalytic activity below the serologic limit of detection. Low residual protein levels are an expected property of the so-called TetON DAS-mediated protein degradation system used due, in part, to the slow kinetics of 'washout' of ATC from bacteria and dissociation of ATC from the Tet repressor required for expression (de-repression) of the protein degradation machinery. In addition, expression of SspB (which drives protein degradation) was also placed under the control of an intermediate, rather than full, strength promoter. We further note that Rv3722c itself, numbers among the top 10-25% of most abundant proteins in *Mtb* (Wang, Proteomics, 2015), adding to the likelihood of slow, if not incomplete, degradation. Moreover, experimental support of this explanation is provided by the demonstration that serial passaging of ATC-deprived bacteria grown in 7H9 (Fig 1A) into fresh 7H9 prevented subsequent growth in absence of ATC. The continued growth of ATC-deprived Rv3722c-TetOn in the absence of serologically detectable levels of Rv3722c is thus the likely consequence of its slow degradation and catalytic activity. These points are now explicitly clarified in the text (line 66-69).*

2. Line 60. It is stressed that the attenuation is seen in GLU-based 7H9 medium. Does this mean that this phenotype is GLU-dependent? What is its phenotype in the absence of GLU?

The reviewer raises an important and interesting question. We found that growth of 3722c-depleted Mtb is attenuated in Glu-containing 7H9, but not in Glu-containing 7H9 supplemented with casamino acids, or a non-Glu-containing, Asn-based media, such as Sauton's. Given our discovery of Glu as a product/substrate of Rv3722c, the ability of casamino acids to rescue growth, and Asp as a putatively matching in vitro product/substrate pair for Rv3722c, we initially hypothesized that either Glu was toxic to 3722c-depleted Mtb or that 3722c-deletion resulted in a strict and specific auxotrophy for Asp (that could be rescued by Asn). However, we observed that Rv3722c-deficient Mtb grew like wildtype in Sauton's media supplemented with Glu at concentrations contained in 7H9, or in Sauton's media in which Asn was replaced with Glu. We further observed that Rv3722c-deficient Mtb conversely failed to grow in 7H9 supplemented with Asn at concentrations contained in Sauton's, or in 7H9 in which Glu was replaced with Asn. Rv3722c-deficient Mtb also failed to grow in a minimal salts media (TSM), which supported growth of wild type Mtb and contained ammonia, rather than Glu, as sole nitrogen source. These experiments thus show that the growth attenuation observed in 7H9 media is NOT strictly Glu-dependent nor strictly driven by the nitrogen source in the media. We have now clarified this in the text (line 76-82).

3. Figure S1 shows that Rv3722c Tet-On grows like WT in Sauton's medium. Does this mean that Asparagine rescues the phenotype or is it the absence of glutamate? This should be worked out.

Neither the absence of Glu, nor the presence of Asn was responsible for the growth phenotype in Sauton's, as compared to 7H9 (see answer to previous question). In additional experiments in which we systematically swapped individual components of 7H9 and Sauton's from one media into and out of the other, we were unable to identify a single component capable of explaining the differential growth phenotype (as detailed in response to comment 11 below).

4. The mouse experiment in figure 2B and its repeat experiment in Figure S2 are confusing.

Figure 2B clearly shows no growth of pre-depleted Rv3722-TetOn in mouse lungs. However, Figure S2 shows that Rv3722-deficient MTB (ATC-responsive) are multiplying in the lungs to a burden of 10⁴. Then at later time points this population is taken over by escape "mutants" that were non-responsive to ATC. There are several issues and questions that arise from these two experiments.

a. Reproducibility of experiment 2B is not demonstrated. In fact, in the repeat experiment, pre-depleted Rv3722-TetOn can grow to a burden of more than 10000 per lung even in the absence of doxycycline. Then the population is taken over by escape mutants.

We believe that the differences seen in Fig 2B and Fig S2 reflect differences in the level of Rv3722c pre-depletion that was achieved in each experiment, and lie below the limit of detectability by Western blotting. As detailed in our response to Comment 1, depletion of Rv3722c is both slow and incomplete, owing, in part, to the design of the proteolytic degradation system used and the apparent biological abundance/excess of Rv3722c itself. As astutely pointed out below, it is possible that differences in inoculum may have also contributed to this inter-experimental variation. Experimental differences notwithstanding, the main and shared endpoint of these 2 experiments, is the finding that Rv3722c is essential for establishing an infection in vivo, which is fundamentally reproduced in both experiments.

b. Given these discrepancies the normalization in Figure 2B should be removed and the real numbers shown as this might hide a possible reason for this discrepancy. As stated in the figure legend, the number of Rv3722-TetOn was lower than that of wildtype. A lower burden at 24 hours could explain why escape-mutants do not arise in Figure 2B while in Figure S2 they arise after 3 weeks. However, this still cannot explain why in one experiment pre-depleted Rv3722-TetOn multiplies in the first 2 weeks and in the other it doesn't.

We thank the reviewer for pointing this out and have changed the normalized data in Fig 2B to absolute numbers. For transparency, we have also changed Fig 2B and S2 to display individual data points instead of means with standard deviation. We finally split Fig S2 into one panel for ATC-dependent bacteria and another panel for non ATC-dependent mutants. The difference in initial growth between the two experiments is addressed in our answers to the preceding question.

c. The nature of the reversion is not revealed in the manuscript. This should be shown and discussed.

Given the high frequency of reversion that we and others have observed using Tet-responsive repressors, mutations in the regulatory system (TetR or sspB) are most likely responsible for phenotypic reversion (Boldrin, Scientific Reports, 2019). Unfortunately, no lung homogenates remain to explicitly confirm this. Nevertheless, we would like to stress that the main point here is that the bacteria that ultimately established an infection were phenotypically non ATC-dependent, regardless of the exact mechanism, and that our findings focus exclusively around ATC-dependent phenotypes.

d. It is unclear what non-ATC responsive means (see figure S2). If Rv3722-TetOn is not ATC responsive anymore, it means it should not grow on plates without ATC. However, the opposite is the case, Rv3722-TetOn even grows on plates without ATC. Maybe ATC-dependent/independent (as used in Fig 2B) is the better choice.

We agree that the term non ATC-responsive is unclear and changed it to non ATC-dependent as suggested throughout.

e. It is common in the field that not only lung burden but also burden in at least one other organ are shown (usually spleen).

We agree that showing the bacterial burden in another organ is useful and have added spleen data to Fig 2. These data are similar to the lung data and corroborate the essentiality of Rv3722c in mice.

5. All in vitro growth curve experiments 1A, 1B, 1F, S1, 7B, don't seem to have error bars.

Error bars are now included in all these figures, but in most cases the small error bars are smaller than the marks that represent the mean. As a resource to the readers, we have included the source data in the revised manuscript.

6. In Figure 3A it is not entirely clear what is meant by "features related to" the feature with m/z 144.05 means. Do you mean fragments?

We apologize for the confusion and have specified how these features relate to the main feature in the legend of Fig 3: "red dots represent features related (fragments, adducts, dimers and isotopes) to ketoglutaramate (m/z 144.03 [M-H]-)". The specific features are listed in the Source Data file.

7. Based on the biochemical experiments it is clear that Rv3722c is the dedicated aspartate aminotransferase, yet no attempts were made to get the structure with Asp bound?

We thank the reviewer for this question. Unfortunately, despite concerted efforts, attempts to co-crystallize or soak preformed crystals with aspartate were unsuccessful. This may, in part, be explained by the fact that Asp is the main substrate for the enzyme, and that it was rapidly converted into product oxaloacetate – which is unstable in aqueous environments due to spontaneous decarboxylation to pyruvate. Attempts to similarly co-crystallize the enzyme with the aspartate analog, α -methyl aspartate did not yield crystals suitable for data collection. Soaking the analog into preformed crystals led to rapid loss of resolution – thus making the data unsuitable for structure determination. We have now added a sentence stating our attempts to generate a structure with Asp bound (line 233-234). Nevertheless, glutamate proved to be a reliable substrate as we were able to co-crystallize it with the enzyme – thus allowing us to explain the mode through which dicarboxylic substrates bind to Rv3722c.

8. Metabolomics methods section: quenching in cold PBS is not specific enough. What was the temperature of PBS, how were the cells washed (centrifuge speed, temperature and time)?

Bacteria were washed with ice-cold PBS, spun down for 5-10 minutes at 3,000 g and 4 °C. We have now added this information to the method section.

9. Media preparation: was the medium pH adjusted when 3 mM amino acids were added? Certain amino acids at such high concentrations, change the pH of 7H9 drastically.

The pH of the prepared media was checked and adjusted when necessary.

10. Is it possible that due to the high glutamate concentration in the 7H9 medium the Rv3722-depleted cells have a much lower aKG level and would need aKG supplementation in addition to aspartate in order to fully complement the growth inhibition? Indeed, looking at Figure S15, addition of Casein shows a significant increase of aKG.

We thank the reviewer for raising this possibility. It is indeed true that the other product of the Asp-producing reaction is aKG. We have performed additional growth experiments to test this hypothesis and found that aKG alone and in combination with Asp did not rescue the growth defect of Rv3722c-deficient Mtb (see below).

11. Why was Sauton's medium not used to find the growth complementing component? In Sauton's medium Rv3722-depleted cells grow like WT, but it is not clear why? Given the limited amount of media components in Sauton's it would be easier to determine which component rescues Rv3722-depleted cells in this medium.

We thank the reviewer for asking. Our thinking was along the same lines, and we were optimistic that we would find a media component that could explain the difference in growth between 7H9 and Sauton's. Interestingly, we also found that Rv3722c was essential in a minimal media that was very similar to Sauton's (see below).

For 1 L of culture medium	7H9 Essential	Sauton's Not essential	Minimal media Essential
Ammonium sulfate	0.5 g		
L-Glutamic acid	0.5 g		
L-Asparagine	-	4 g	0.5 g
Sodium Citrate/ Citric acid	0.1 g	2 g	
Glycerol	0.2%	6%	0.2%
Ferric ammonium citrate	0.04 g	0.05 g	0.05 g
Magnesium sulfate	0.05 g	0.5 g	0.5 g
Calcium Chloride	0.5 mg		0.5 mg
Zinc Sulfate	1.0 mg	1.0 mg	0.1 mg
Copper Sulfate	1.0 mg		

Disodium phosphate	2.5 g		2.5 g
Monopotassium phosphate	1.0 g	0.5 g	1.0 g
Pyridoxine	1 mg		
Biotin	0.5 mg		
Albumin	5 g		
Glucose	2 g		
Sodium Chloride	0.85 g		
Tyloxapol	0.04%		0.2%
Tween 80		0.05%	

As explained in our response to Comment 3 above, swapping Glu and Asn in 7H9 and Sauton's, or adding Asn or Glu to 7H9 and Sauton's, did not change the growth characteristics. Additional experiments in which we systematically interchanged 8 additional single media components between 7H9, Sauton's and the minimal media did not reveal major growth changes either. Taken together, these negative results have led us to conclude that the phenotype of Rv3722c is the result of a complex combination of media components.

Reviewer #3 (Remarks to the Author):

The manuscript titled "Rv3722c governs aspartate-dependent nitrogen metabolism in Mycobacterium tuberculosis" by Jansen et al., is an effort in the identification of the previously misrecognized/underrecognized role of Rv3722c as an aspartate aminotransferase of Mycobacterium tuberculosis. Although the manuscript is reasonably well-written and the results are nicely presented using several high-tech methods including TetON system, metabolomics, and structural biology, there are three major issues.

Major comments;

1. Rv3722c is already annotated as an aminotransferase of Mtb with the X-ray structure possessing an aspartate aminotransferase domain (<https://www.ebi.ac.uk/pdbe/entry/pdb/5C6U>), although the authors have added the multifaceted essentiality of Rv3722c to this manuscript. Thus, the rationale for Rv3722c should be reconsidered in page 3, lines 40-44 in introduction section. In the same context, the authors should describe the reasons more clearly for how and why the authors chose and investigated rv3722c among many unrecognized/unidentified genes in Mtb in introduction section. *We thank the reviewer for pointing out this apparent inconsistency, and are happy to clarify. The structure of Rv3722c (5C6U) was solved by our group in collaboration with the Midwest Center for Structural Genomics. At the time the structure was deposited into the PDB, the only thing we knew was its class-level annotation as an aminotransferase, and that it had a type I fold of PLP-binding enzymes. This, however, did not tell us anything about its function. In fact, in 2016, activity-based protein profiling implicated Rv3722c as a serine hydrolase (Ortega, Cell Chem Biol, 2016).*

Confusingly, PLP-binding enzymes are interchangeably referred to either by the name of the class to which they belong (e.g. fold-type III) or by the name of the founding member of that group (e.g. Alanine racemase family) (Eliot and Kirsch, 2004). For example, all members of group type III are also known as members of the alanine racemase family. This does not mean all members of this group are alanine racemases or even other racemases. In fact, the group is dominated by L-amino acid decarboxylases.

Likewise, the fold-type I PLP-dependent enzymes (as a class), are also referred to as the family of aspartate aminotransferases. The presence of an aspartate aminotransferase domain thus only implicates a protein as a member of the family of aspartate aminotransferases/fold-type I PLP-dependent enzymes, and not necessarily as an actual AspAT. The family of aspartate aminotransferases/fold-type I PLP-dependent enzymes actually harbors a wide variety of proteins that have enzymatic functions ranging from different types of aminotransferases, to hydroxymethyltransferases, ligases, lyases, and decarboxylases (Grishin, Protein Science, 1995) and even transcription factors (Bramucci, Biochemical and Biophysical Research Communications, 2011). Taken together, the enzymatic function of Rv3722c was unclear at the start of this study.

The PDB entry 5C6U was already mentioned in the method section because it was used as a template in phase determination during structure solution of the enzyme bound to substrates (“Protein crystallization and Structure determination”). We also mentioned it when comparing the effect of residue conformational change that results upon binding of L-kyn to the enzyme (Fig S10).

In the revised version (line 42-47), we have now added a section to the introduction in which we mention 5C6U and explain that class I PLP-binding proteins are also referred to as the family of aspartate aminotransferases. Since 5C6U is not ligand-bound, and since our overall aim was to provide structural basis of dual substrate recognition, we elected not to include it as we deemed it redundant.

We chose to work on Rv3722c because it is essential, abundant, poorly annotated, and because it contains a PLP-binding fold and thus likely serves as an enzyme. We have added these reasons to the introduction (line 48-50).

2. In Figure 1, Figure 2, Figure S1, and Figure S2, Rv3722c-TetOn seems essential for the Mtb’s growth in 7H9 broth but is dispensable in Sauton’s medium. In particular, results of in vivo experiments are intriguing. However, the authors should investigate the opposite way in vivo experiments between Figure 2B and Figure S2. For example, the authors proved in vitro essentiality of Rv3722c by comparing the growth of Rv3722c-TetOn with/without ATC between two different media. With these results, the authors should investigate in vivo essentiality of Rv3722c-TetON cultured from 7H9 in mice with ATC (Doxycycline) in Figure 2B. In the same line, the initial infectious doses are quite different between Figure 2B and Figure S2, although the authors displayed Log10 normalized data in Figure 2B. Thus, I suggest that the author perform the experiment of Figure 2B with a high infectious dose (>1,000 initial dose or more) to prove whether this gene is virtually essential in vivo.

We report data from two independent in vivo experiments, one in which wildtype Mtb was compared to the mutant strain pre-depleted in Sauton’s (Fig 2B) and one in which the mutant strains (pre-depleted or not in Sauton’s) were used to infect mice that were then fed regular chow or doxycycline-containing chow, respectively (Fig S2). Despite these differences as well as differences in the inocula used (discussed below), both experiments clearly show that Rv3722c is essential for Mtb to establish an infection in vivo.

We agree that testing the Rv3722c-proficient mutant (pre-cultured in 7H9 with ATC) in mice on a control or doxycycline-containing diet would be interesting, but also believe that the reported data (described above) clearly demonstrate the essentiality of Rv3722c for the establishment of Mtb infection in vivo. While the setup proposed by the reviewer could address the essentiality of Rv3722c in the maintenance of a chronic infection, which we also agree would be interesting, we respectfully believe that such an experiment lies beyond the scope of the current work, which focuses on the biochemical function of Rv3722c, and is instead better suited for follow-on studies that would benefit from the development of strains capable of more rapidly depleting Rv3722c in an inducible manner, and are ongoing. As such, we believe that performing such an experiment with our existing strains would not add further to the demonstrated essentiality of Rv3722c in the establishment of Mtb infection in vivo.

We have now changed the log10-normalized scale in Fig 2B to a scale with absolute CFUs, to allow a comparison between the inocula of Fig 2B and Fig S2. For transparency, we have also changed Fig 2B and S2 to display individual data points instead of means with standard deviation. We finally split Fig S2 into one panel for ATC-dependent bacteria and another panel for non ATC-dependent mutants.

The reviewer rightfully points out that the inoculum size of the mutant strain in Fig 2B is relatively low. We were thinking along the same lines and thus performed the second infection experiment presented in Fig S2, which resulted in an inoculum that was approximately two orders of magnitude greater but was associated with an increased number of pre-existing non ATC-dependent mutants that were able to establish a productive infection (Fig S2B). Moreover, the complete overgrowth of ATC-dependent mutants by non ATC-dependent mutants provides clear evidence of essentiality/fitness requirement of Rv3722c for establishment of Mtb infection in vivo.

3. The authors very nicely described the procedures of metabolomics techniques such as TOF MS and Q-TOF to identify

the metabolites in pages 28 - 29 in Materials and Methods. However, it would be highly helpful to the readers if the authors add information how to get ID of metabolites such as retention time and spectrum data for each metabolite as supplementary tables. For example, it is helpful to provide comparison data for retention time (RT) with standard in TOF MS. For another example, please provide transition values to figure out ion fragment pattern and RT in Q-TOF analysis. *For the identification of metabolites that we routinely detect, we rely on retention time in combination with accurate mass data that were previously validated by fragmentation spectra. The accurate mass of any metabolite is a function of its chemical formula and can be calculated using free online tools. As a helpful resource for readers, we have now added information on the retention time of the reported metabolites in the source data file. Although we did not make use of fragmentation for the detection of any of the metabolites except for ketoglutaramate (Fig 3C), the free online metabolite database Metlin (https://metlin.scripps.edu/landing_page.php?pgcontent=mainPage) contains fragmentation spectra of over 500,000 compounds. These spectra were obtained using an MS/MS system that is almost identical to the one used in this study.*

Minor comments;

1. In line 135, 2,5 h -> 2.5 h?

Done

2. C8 and Rv3722c are co-transcribed by RT-PCR (Arnvig and Young, 2009). Also, DNA polymerase III subunit dnaZX (Rv3721c) may lie in the same operon? Please discuss the downstream effect of Rv3722c-TetON if those genes are affected.

*C8 lies upstream of Rv3722c and should therefore not be affected by the construct. The controlled proteolysis system relies on introduction of a DAS-tag at the C-terminus of the target gene. The DAS-tag cassette contains a *hygR* promoter, which drives transcription of downstream genes, in this case Rv3721c. Even if transcription of the downstream Rv3721c is affected, it will be affected equally in the presence or absence of ATC, because levels of Rv3722c are controlled at the protein level (by controlled proteolysis) and not at the transcription level.*

3. In lines 436-439 in page 25, the authors suggest Asp biosynthesis may be a new potential drug target in Mtb. However, Mtb goes through the latency, so what if Rv3722c and the relevant process facilitate the latency? If so, this may lead to more drug tolerance. For example, Figure 2A, Rv3722c-TetON is more resistant in presence of IFN-gamma inside macrophages because there is no difference in Rv3722c-TetON growth regardless of IFN-gamma. Thus, I am wondering drug susceptibility of the Rv3722c-TetON to INH and RIF.

We thank the reviewer for raising the role of Rv3722c in latency. We performed additional experiments to test the effect of Rv3722c on the susceptibility to isoniazid and rifampicin and observed no difference in MICs.

4. Throughout the manuscript, the authors statistically used unpaired Student's t-test by representing mean-/+ SD. However, the sample size is not sufficient to follow this statistic analysis. Thus, please use Mann-Whitney test as a nonparametric test.

We thank the reviewer for pointing out the inability of the unpaired Student's t-test to provide a statistically meaningful p-value with a per group sample size of three. We agree that the p-values associated with such comparisons are not robust due to their undue vulnerability to the impact of potential outlier values. We therefore sought the consultation of academic statisticians who advised us to apply the alternative nonparametric Mann-Whitney U-test, as suggested by the reviewer. However, we note that, due to the same sample size limitation, this test is inherently unable to achieve p-values smaller than 0.1 with a per group sample size of 3 – no matter how big the difference between groups. Notwithstanding this limitation, our statisticians recommended the use of the Mann-Whitney U test to both provide a statistically defined measure of significance and serve as an independent sensitivity analysis for outliers that may have unduly affected/skewed the reliability of the t-test. Accordingly, we applied a combination of both a two-sided unpaired Student's t-test ($p < 0.05$) and Mann-Whitney U rank testing ($p\text{-value} = 0.1$, implying full rank separation) to help mitigate this sample size limitation. We have now added a section "Statistical methods" to the manuscript, in which we describe the combined use of Student's t-test and the Mann-Whitney U-test, to define statistically meaningful differences. In addition, we explicitly report the means and standard deviations of each reported group ($n=3$) in the figure, with individual values in the Source Data.

In addition, the number of mice is too small to conclude in vivo testing.

Animal studies were performed using 5 mice per group to ensure statistical significance. This is based on power calculations to detect a difference of at least 1 log₁₀ in CFU between two strains ($\alpha = 0.05$, power = 95%), in lungs and spleens.

5. In Figure 1, please present virtual CFUs rather than O.D. value.

We respectfully disagree and believe that the OD value is a better way to represent the data because it represents the primary data, because the relationship between OD and CFUs is dependent on factors such as cell size which makes calculation of virtual CFUs unreliable, and because it is common practice in the field.

6. In line 86 of page 6, macrophages are not in vivo experiment. I am wondering the initial CFUs in macrophages are same between Figure 2A experiments in absence/presence of IFN-gamma? How dose nitric oxide affect the growth of WT and Rv3722c-TetON in macrophages by treating IFN-gamma? In Figure 2B, 0 is correct even though the authors are expressed the normalized data?

We have removed "in vivo" in reference to the macrophage infection experiments.

The data presented in Fig 2A were collected from the same experiment, using the same bacterial culture. The initial CFUs in both experiments were therefore identical.

Nitric oxide is indeed one of the main, though not exclusive, downstream bactericidal effectors of IFN-gamma. Although we agree that the question raised by the reviewer is fascinating and deserves further research, we respectfully think that it is beyond the scope of the current work.

The data in the original version of Fig 2B was presented as the Log₁₀ of the data normalized to 24h. Per definition, the normalized value at 24h is 1, which corresponds to Log₁₀ value of 0. In the revised version, however, we have changed the normalized scale to a scale that represents absolute CFUs.

7. In line 461, Pmel -> italic

Done

8. In line 573, clarify LCM and L cell.

We have now clarified L cells are L929 cells, and added a reference describing their use as a source of macrophage colony-stimulating factor for the generation of primary bone marrow-derived macrophages (line 612).

REVIEWERS' COMMENTS:

Reviewer #2 (Remarks to the Author):

The authors have responded to all my question satisfactorily. It is puzzling that, despite considerable experimental effort, the influence of media components on growth rescue of the mutant could not be disentangled. Nevertheless, even though this would be important information, the manuscript holds a substantial amount of novel findings that are important for the TB research community.

Reviewer #3 (Remarks to the Author):

The reviewer believe that the authors appropriately addressed all raised concerns regarding animal studies and inconsistent citations for the function of Rv3722c. Thus, I do not have any further concerns.